DOI: 10.1038/s41467-017-00791-2　　**OPEN**

# Peroxiredoxin 6 mediates Gαi protein-coupled receptor inactivation by cJun kinase

Selena S. Schattauer[1], Benjamin B. Land[1], Kathryn L. Reichard[1], Antony D. Abraham[1], Lauren M. Burgeno[1,2], Jamie R. Kuhar[1], Paul E.M. Phillips [1,2], Shao En Ong[1] & Charles Chavkin [1]

Inactivation of opioid receptors limits the therapeutic efficacy of morphine-like analgesics and mediates the long duration of kappa opioid antidepressants by an uncharacterized, arrestin-independent mechanism. Here we use an iterative, discovery-based proteomic approach to show that following opioid administration, peroxiredoxin 6 (PRDX6) is recruited to the opioid receptor complex by c-Jun N-terminal kinase (JNK) phosphorylation. PRDX6 activation generates reactive oxygen species via NADPH oxidase, reducing the palmitoylation of receptor-associated Gαi in a JNK-dependent manner. Selective inhibition of PRDX6 blocks Gαi depalmitoylation, prevents the enhanced receptor G-protein association and blocks acute analgesic tolerance to morphine and kappa opioid receptor inactivation in vivo. Opioid stimulation of JNK also inactivates dopamine D2 receptors in a PRDX6-dependent manner. We show that the loss of this lipid modification distorts the receptor G-protein association, thereby preventing agonist-induced guanine nucleotide exchange. These findings establish JNK-dependent PRDX6 recruitment and oxidation-induced Gαi depalmitoylation as an additional mechanism of Gαi-G-protein-coupled receptor inactivation.

[1] Department of Pharmacology, University of Washington School of Medicine, Seattle, WA 98195, USA. [2] Department of Psychiatry, University of Washington School of Medicine, Seattle, WA 98195, USA. Correspondence and requests for materials should be addressed to C.C. (email: cchavkin@u.washington.edu)

Mu opioid analgesics are routinely used for the treatment of severe pain, but the profoundly adverse side effects caused by sustained opioid administration strongly limit their safety and clinical utility[1]. In addition, preclinical and human studies suggest that kappa opioid receptor (KOR) antagonists may have therapeutic utility in the treatment of mood disorders and drug addiction[2, 3], however, selective KOR antagonists have unexplained pharmacological properties, including a long duration of action, that limit their utility[2, 4–6]. Opioid receptor activation stimulates canonical membrane delimited Gβγ regulation of ion channels resulting in analgesia, but also activation of mitogen activated protein kinases (MAPK) including ERK, p38 and cJun N-terminal kinase (JNK) that result in transcription factor phosphorylation and changes in synaptic structural plasticity that may underlie the enduring actions of opioids[7]. Selective activation of specific signaling pathways underlies the evolving concept of ligand-directed signaling, which holds great promise for guiding the development of safer, more functionally selective opioid medications[8, 9]. Advancing these therapeutic concepts requires that we develop better understanding of the ligand-directed signaling mechanisms activated by these opioid drugs.

One form of signaling by G-protein-coupled receptors (GPCRs) involves the activation of arrestin, which sterically inhibits Gβγ activation and promotes MAPK activation. While efficacious opioid agonists including fentanyl, DAMGO, and etorphine activate the typical G-protein receptor kinase (GRK) arrestin-dependent mu opioid receptor (MOR) desensitization, morphine causes arrestin-independent acute analgesic tolerance by a JNK-dependent mechanism[5, 10]. Similarly, selective KOR antagonists paradoxically have collateral agonist activity, resulting in receptor-dependent JNK activation to cause long-lasting KOR inactivation by an arrestin-independent mechanism[4–6, 10]. Pharmacological inhibition or genetic deletion of specific JNK isoforms blocks acute analgesic tolerance to morphine and prevents long-lasting KOR antagonism, as does pretreatment with KOR antagonists which do not activate JNK[4, 5], but how JNK inactivates opioid receptor signaling is not known. These results suggest that JNK may phosphorylate a component of the opioid receptor signaling complex, thereby preventing G-protein activation in an alternative process to arrestin-mediated desensitization.

In this study, we use proteomics to identify peroxiredoxin 6 (PRDX6) as a JNK-regulated mediator of opioid receptor desensitization. We find that opioid and dopamine receptors recruit PRDX6, promoting the generation of reactive oxygen species. This results in depalmitoylation of Gαi and receptor desensitization. Finally, we show that this JNK-PRDX6 pathway results in drug tolerance in vivo.

## Results

**Opioid receptor-Gαi association is JNK regulated.** To define this GPCR-inactivation mechanism, we first determined if the JNK substrate was a membrane-associated component of the receptor signaling complex (Supplementary Fig. 1a). Both the selective mu agonist DAMGO and the kappa agonist U69,593 stimulated [$^{35}$S]GTPγS binding in spinal cord membranes treated with ATP alone or JNK in the absence of ATP (Supplementary Fig. 1b, c). However, treatment with JNK in the presence of ATP inhibited DAMGO-stimulated [$^{35}$S]GTPγS binding to MOR by 60% and completely blocked U69,593 stimulated [$^{35}$S]GTPγS binding to KOR (Supplementary Fig. 1d). Although these results suggest that JNK phosphorylates a component of the opioid receptor signaling complex, which inhibits nucleotide exchange, we were not able to detect direct

JNK-mediated phosphorylation of either the opioid receptors or the G proteins (Supplementary Fig. 1e, f).

An alternative explanation is that JNK phosphorylation recruits an arrestin-like substrate within the signaling complex that occludes receptor G-protein interaction, and we next used quantitative SILAC (stable isotope labeling of amino acids in cell culture) proteomics[11] to determine whether a protein with arrestin-like properties showed increased association with the opioid receptor after JNK activation with norBNI, a selective KOR antagonist with collateral agonist activity at the JNK pathway[4]. Using this approach, we identified proteins associated with myc-tagged KOR in two independent replicates in SILAC labeled HEK293 cells (Supplementary Fig. 2a). Specifically interacting proteins were identified as enriched relative to a control having excess myc peptide as a competitor (log2 value > 0.5 relative to competition control and P-value < 0.05 with reverse media labeling for significance B, an outlier significance score[12]). A total of 65 mycKOR-interacting proteins were identified using mass spectrometry (Supplementary Data 1). To determine the effect of norBNI on these interactions, the ratio of protein interactors detected after norBNI treatment relative to vehicle treatment was then compared (Fig. 1a, Supplementary Data 1). The majority of protein interactions were unaltered by norBNI treatment, and no protein associations were significantly decreased. Although this analysis failed to detect an arrestin-like protein whose association with mycKOR was increased by JNK activation, surprisingly the associations of Gαi and Gβ with mycKOR were significantly increased by norBNI treatment (40% and 100%, respectively).

To validate the SILAC proteomic results, we resolved mycKOR immunoprecipitates on SDS PAGE gels and immunoblotted for Gαi (Fig. 1b). Western analysis confirmed that norBNI treatment significantly increased (48 ± 12%) Gαi immunoreactivity (IR) associated with mycKOR in a JNK-dependent manner, as this increase was not detected in extracts from norBNI treated cells pretreated with the JNK inhibitor SP610025 (Fig. 1b, Supplementary Fig. 3). Treatment with naloxone (a non-selective opioid antagonist that does not promote JNK phosphorylation[6]) did not significantly change Gαi association with mycKOR (Fig. 1b). Furthermore, in the reciprocal experiment in which HEK293 cells stably expressing FLAG-Gαi3 and mycKOR were treated with norBNI prior to immunoprecipitation with anti-FLAG agarose, norBNI increased KOR immunoreactivity detected in the FLAG immunoprecipitate (87 ± 24%). This increase in KOR immunoreactivity was also blocked by SP610025 pretreatment (Fig. 1c).

In our prior studies[4–6, 10], JNK activation was inferred by the detection of phospho-JNK-ir by western blots and by sensitivity to SP610025 and JNK1 gene knockout. In the present study, norBNI activation of JNK was directly shown by measuring phosphorylation of the JNK substrate cJun (Supplementary Fig. 3). The role of JNK was corroborated by showing that cJun phosphorylation was blocked by the JNK inhibitors JNK-IN-8 or SP610025, and the selectivity of both JNK-IN-8 and SP610025 in our assays was verified by showing that neither inhibitor affected ERK phosphorylation (Supplementary Fig. 3b).

These results demonstrate that norBNI treatment increases Gαi association with KOR in a JNK-dependent manner. To determine if this was specific to KOR or a more general mechanism employed by other GPCRs, we treated mycMOR expressing HEK293 cells with morphine prior to immunoprecipitation. An increase in Gαi association (182 ± 74%) was also observed in mycMOR immunoprecipitates that was SP610025-sensitive (Fig. 1d). From these results, we conclude that JNK activation by sustained norBNI or sustained morphine increases the Gαi-receptor association. How this increased association occurs and how it may impede receptor signaling were not clear, because an

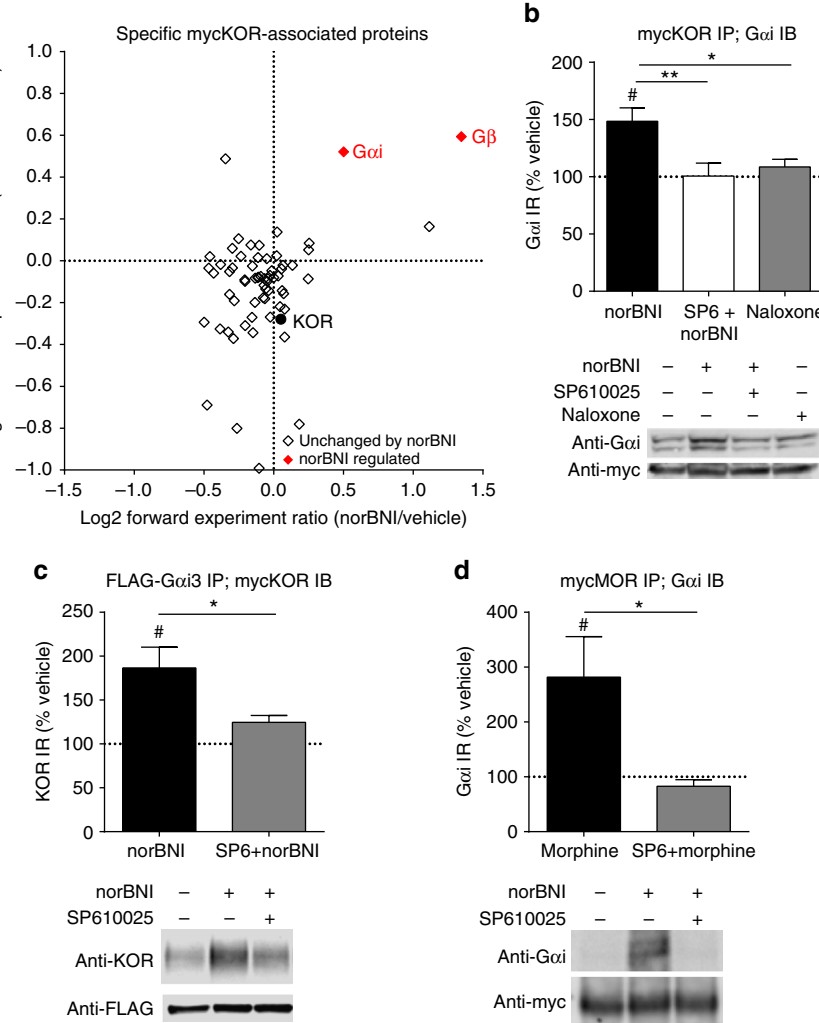

**Fig. 1** NorBNI treatment increases KOR-Gαi association in a JNK-dependent manner. **a** Scatterplot of mycKOR-interacting proteins, depicting the ratio of norBNI/vehicle of the forward and reverse replicate. For experiment design, see Supplementary Fig. 2a; for individual proteins, see Supplementary Data 1. Proteins increased by norBNI in both replicates are displayed in the *upper right quadrant*, and proteins whose associations were decreased by norBNI treatment in both replicates are shown in the *lower left quadrant*. Proteins whose associations were significantly changed by norBNI treatment (norBNI/vehicle log2 ratio > 0.5 and significance B *P*-value < 0.05 in both replicates) are indicated in *red*. **b** NorBNI (10 μM) significantly increased Gαi immunoprecipitated with myc in mycKOR-expressing cells (#, one sample *t*-test, correction for multiple comparisons); this increase was blocked by 1 μM SP610025 and was not observed following 10 μM naloxone (one-way ANOVA, *P* < 0.01, *n* = 8-13; *P* < 0.05, **P* < 0.01, Holm–Sidak post hoc analysis). **c** NorBNI significantly increased KOR immunoprecipitated with FLAG in mycKOR and FLAG-Gαi3 expressing cells (#, one sample *t*-test, correction for multiple comparisons), and was blocked by SP610025 pretreatment (*, Student's *t*-test, *P* < 0.05, *n* = 6-7). **d** Morphine (10 μM) significantly increased Gαi immunoprecipitated with myc in mycMOR-expressing cells (#, one sample *t*-test, correction for multiple comparisons) and was blocked by SP610025 pretreatment (*, Student's *t*-test, *P* < 0.05, *n* = 4-7). *Error bars* represent mean ± SEM

increase in G-protein association would be expected to increase signaling.

**Mass spectrometry reveals PRDX6-Gαi association**. To assess how JNK activation affected Gαi associated proteins, we next employed the SILAC proteomic approach using HEK293 cells stably expressing FLAG-Gαi3 with mycKOR (Supplementary Fig. 2b). This approach identified a total of 76 proteins that were specifically associated with FLAG-Gαi3 (Fig. 2a, Supplementary Data 2). While the majority of proteins interacting with FLAG-Gαi3 were unchanged by norBNI treatment, KOR association was increased by 40% (Fig. 2a), consistent with the prior SILAC and co-immunoprecipitation results (Fig. 1). We also identified peroxiredoxin 6 (PRDX6) as FLAG-Gαi3 associated protein specifically enhanced by norBNI treatment (Fig. 2a). PRDX6

is a bifunctional enzyme, with separately regulated glutathione peroxidase activity and phospholipase A2 (PLA2) activity[13]. However, a relationship between PRDX6 and opioid receptor signaling has not been previously reported.

Prior studies have shown that phorbol esters or the Gαq-coupled angiotensin receptor agonist-stimulated MAPK phosphorylates PRDX6, leading to increased membrane localization and selective stimulation of its PLA2 activity[14, 15]. To determine whether JNK-activated PRDX6 was responsible for the enhanced interaction between KOR and Gαi, we used MJ33, which selectively inhibits the phospholipase activity of PRDX6 and pancreatic (type IB) PLA2, without affecting cytosolic (type IV) PLA2 or PRDX6 peroxidase activity[14–18]. We found that MJ33 blocked the norBNI-induced increase in Gαi immunoreactivity co-precipitated with mycKOR (Fig. 2b). MJ33 pretreatment had no effect on norBNI-stimulated JNK phosphorylation (39 ± 15%

and $31 \pm 16\%$ increase after vehicle and MJ33 pretreatment, respectively; Fig. 2c), demonstrating that PRDX6 activation was downstream of norBNI-activated JNK. We were unable to detect PRDX6 phosphorylation by mass spectrometry following norBNI treatment, however norBNI treatment increased PRDX6 colocalization with KOR, as measured by the correlation between the spatial distribution of immunofluorescence intensity of PRDX6 and KOR immunoreactivity in cells (Fig. 2d–f, Supplementary

Fig. 4a). We further verified that norBNI-stimulated PRDX6 translocation to the membrane ($38 \pm 13\%$) by western analysis of PRDX6 in the membrane fraction of mycKOR expressing HEK293 cells (Fig. 2g, h). These changes were blocked by JNK inhibitors (Fig. 2d–h). In addition, norBNI treatment increased calcium-independent PLA2 activity only in the membrane fraction of cells ($16 \pm 5\%$), which was blocked by MJ33 or the selective JNK inhibitor JNK-IN-8[19] (Fig. 2i, Supplementary

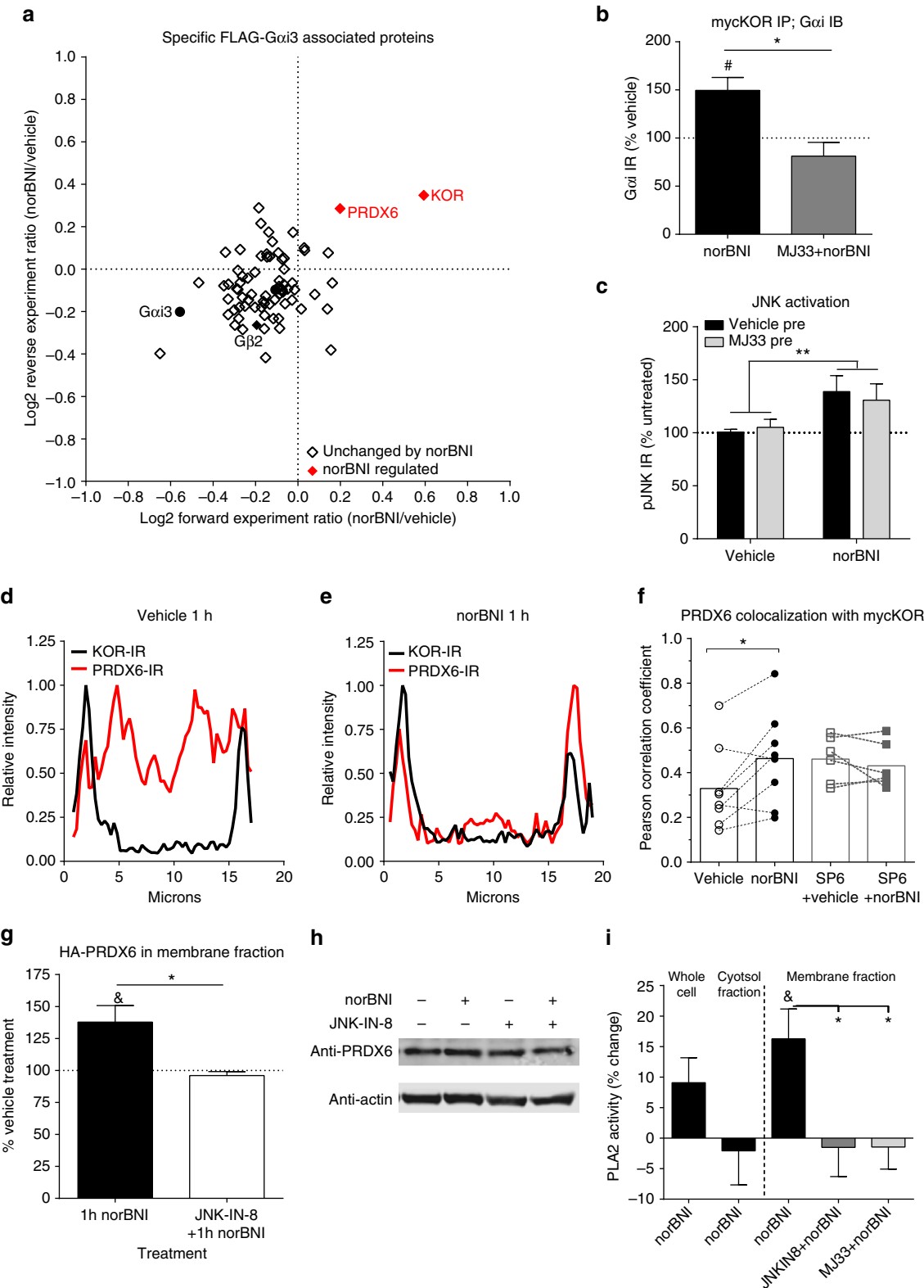

Fig. 3). These findings suggest that JNK activation results in PRDX6 translocation to the plasma membrane where its PLA2 activity increases opioid receptor-Gαi association.

**Opioids promote ROS generation and reduce Gαi palmitoylation.** One established role of PRDX6's PLA2 activity is enhancing NADPH oxidase-mediated production of reactive oxygen species (ROS)[16]. On the basis of this known function, we tested the effect of opioids on ROS production using the CellROX Green detection assay[20] and the genetically encoded ROS sensor, HyPerRed[21] (Fig. 3a–d, Supplementary Fig. 5). CellROX Green is a cell-permeable dye which fluoresces with an absorption/emission of 485/520 nm upon oxidation-induced DNA binding; in contrast to most ROS-sensitive dyes, it is detergent-resistant, formaldehyde-fixable, and stable for up to 24 h. HyPerRed is a fluorescent protein expressed by transfection with an absorption/emission of 575/605 nm; this fluorescence is strongly increased by formation of an oxidation-induced disulfide bond between Cys199 and Cys208. NorBNI increased CellROX Green staining at 60 min with an $EC_{50}$ of 21 nM (Supplementary Fig. 5a, b). HyPerRed fluorescence was also increased by norBNI in a time-dependent manner (Supplementary Fig. 5c, d). The norBNI-stimulated increase CellROX Green staining in mycKOR HEK293 cells ($130 \pm 21\%$) was blocked by pretreatment with MJ33, SP610025, JNK-IN-8, the nonspecific antioxidant N-acetyl-L-cysteine (NAC), or siRNA knockdown of PRDX6 (Fig. 3a, c, Supplementary Fig. 5e–g). In contrast, naloxone did not stimulate ROS production. NorBNI induction of ROS was KOR-mediated, as norBNI treatment of mycMOR expressing cells had no effect on ROS generation (Fig. 3d). In parallel, morphine treatment increased CellROX Green staining in mycMOR cells ($220 \pm 37\%$), which was blocked by SP610025 (Fig. 3b, d). Fentanyl, an efficacious MOR agonist that stimulates JNK by a GRK/arrestin-dependent mechanism[6], did not increase CellROX Green staining in mycMOR expressing HEK293 cells (Fig. 3d). Together, these results indicate that arrestin-independent JNK activation by norBNI or morphine results in JNK-dependent PRDX6-mediated ROS production, an effect not previously attributed to opioid receptor activation.

ROS can oxidize proteins, particularly at cysteines, resulting in both reversible (sulfenylation) and irreversible (sulfonylation) modifications[22, 23]. How this might affect opioid receptor signaling was not clear, but Gα proteins are reversibly palmitoylated through a thioester bond to a conserved N-terminal cysteine residue, which has been shown to affect membrane association of G proteins[24, 25]. In addition, GPCRs

have a conserved palmitoylated cysteine that regulates their conformation[26]. G-protein palmitoylation is recognized as a reversible regulatory mechanism[24–26], but its role in controlling GPCR signaling is not characterized. We hypothesized that the ROS production induced by norBNI would oxidize cysteine residues on Gαi or KOR, disrupting normal protein palmitoylation, and modifying receptor/G-protein interaction to lock inactive Gαi in the receptor complex.

We used an acyl-biotin exchange assay[27] in conjunction with KOR-Gαi co-immunoprecipitation to test this hypothesis. In this assay, palmitoylated cysteines are converted to biotinylated-cysteines, which can then be quantified by streptavidin binding (Fig. 3e). We found that palmitoylation of KOR was unchanged following norBNI (Supplementary Fig. 6a, b). However, norBNI treatment significantly reduced biotinylation of Gαi associated with mycKOR, indicative of reduced Gαi palmitoylation ($52 \pm 12\%$ of vehicle treated, Fig. 3f, g), which was blocked by MJ33 pretreatment. No change in palmitoylation was observed in the Gαi fraction that was not immunoprecipitated with mycKOR (Supplementary Fig. 6c, d).

**PRDX6 mediates opioid receptor inactivation in vivo.** To determine if PRDX6's PLA2 activity and the resulting generation of ROS regulates opioid receptor function in vivo, we used the warm water tail withdrawal assay to measure opioid-induced antinociception. As previously published[4], a single injection of norBNI 3 days prior to KOR agonist treatment blocked kappa opioid antinociception as measured by an increase in tail flick latency induced by the KOR agonist U50,488 (Fig. 4a). Pretreatment with MJ33 1 h prior to norBNI prevented norBNI inhibition of U50,488 induced antinociception 3 days later. In contrast, AACOCF3, an inhibitor of cytosolic PLA2 with low activity at PRDX6[14], had no effect on norBNI inhibition of U50,488. Consistent with the proposed role of ROS, the long duration of norBNI antagonism was also blocked by pretreatment with the non-selective antioxidant NAC (Fig. 4b). MJ33 did not directly affect the antinociceptive response when given immediately prior to U50,488, indicating that MJ33 blocked KOR inactivation in vivo through inhibition of PRDX6 rather than a direct effect on KOR function (Supplementary Fig. 7a). Together, these data show that the PLA2 activity of PRDX6 and the subsequent generation of ROS are required for KOR inactivation in vivo.

We sought to test whether this mechanism of receptor regulation was shared with MOR. Studies have previously implicated NADPH oxidase in morphine tolerance[28, 29]. To determine the effect of MJ33 on acute tolerance to morphine,

---

**Fig. 2** PRDX6 is recruited to Gαi3 following norBNI treatment and regulates KOR/Gαi association. **a** Scatterplot of FLAG-Gαi3-interacting proteins, depicting the ratio of norBNI/vehicle in forward and reverse replicates. For experiment design, see Supplementary Fig. 2a; for individual proteins, see Supplementary Data 1. Proteins whose associations were changed by norBNI treatment in both replicates are indicated in *red*. **b** The increase in Gαi immunoprecipitated with myc after norBNI in mycKOR expressing cells was blocked by 10 μM MJ33 pretreatment (*, Student's *t*-test, $P < 0.05$, $n = 3$–5). **c** NorBNI treatment (10 μM, 1 h) significantly increased phospho-JNK immunoreactivity in KORGFP expressing cells and was not blocked by 10 μM MJ33 pretreatment. (Significant effect of norBNI, **$P < 0.01$, but not MJ33 or interaction; two-way ANOVA, $n = 9$). **d–f** MycKOR expressing cells were immunostained for KOR and PRDX6. Colocalization was quantified by the Pearson correlation coefficient between intensity of KOR and PRDX6 immunoreactivity across 7–10 cells for each replicate. Representative images are shown in Supplementary Fig. 4a. Representative line plot profile analysis for a vehicle (**d**) and norBNI (10 μM, 1 h) (**e**) treated cell. Line plot is presented as the intensity normalized to the maximal intensity for that cell. **f** NorBNI treatment (significantly increased colocalization of KOR-IR and PRDX6-IR (*, paired *t*-test, $P < 0.05$, $n = 8$). No change after norBNI was observed when pretreated with 1 μM SP610025 (paired *t*-test, $P = 0.6$, $n = 6$). **g** NorBNI (10 μM, 1 h) significantly increased HA immunoreactivity in the membrane fraction of HEK293 cells stably expressing mycKOR and HA-PRDX6 (($P < 0.05$) one sample *t*-test, $n = 7$). This increased was blocked by pretreatment with JNK-IN-8 (1 μM) (Student's *t*-test, $P < 0.05$, $n = 4$). **h** Representative immunoblot for **g**. **i** HEK293 cells stably co-expressing mycKOR and HA-PRDX6 were treated with vehicle or norBNI (10 μM, 1 h) and lysed in PLA2 buffer. Whole cell, membrane, or cytosol fractions were collected and analyzed for PLA2 activity. NorBNI selectively increased PLA2 activity in the membrane fraction (& ($P < 0.05$) one sample *t*-test, $n = 8$). This increase was blocked by pretreatment with JNK-IN-8 (1 μM) or MJ33 (10 μM) ($P < 0.05$, one-way ANOVA; *$P < 0.05$ Holm–Sidak post hoc vs. vehicle + norBNI; $n = 4$-8). *Error bars* represent mean ± SEM

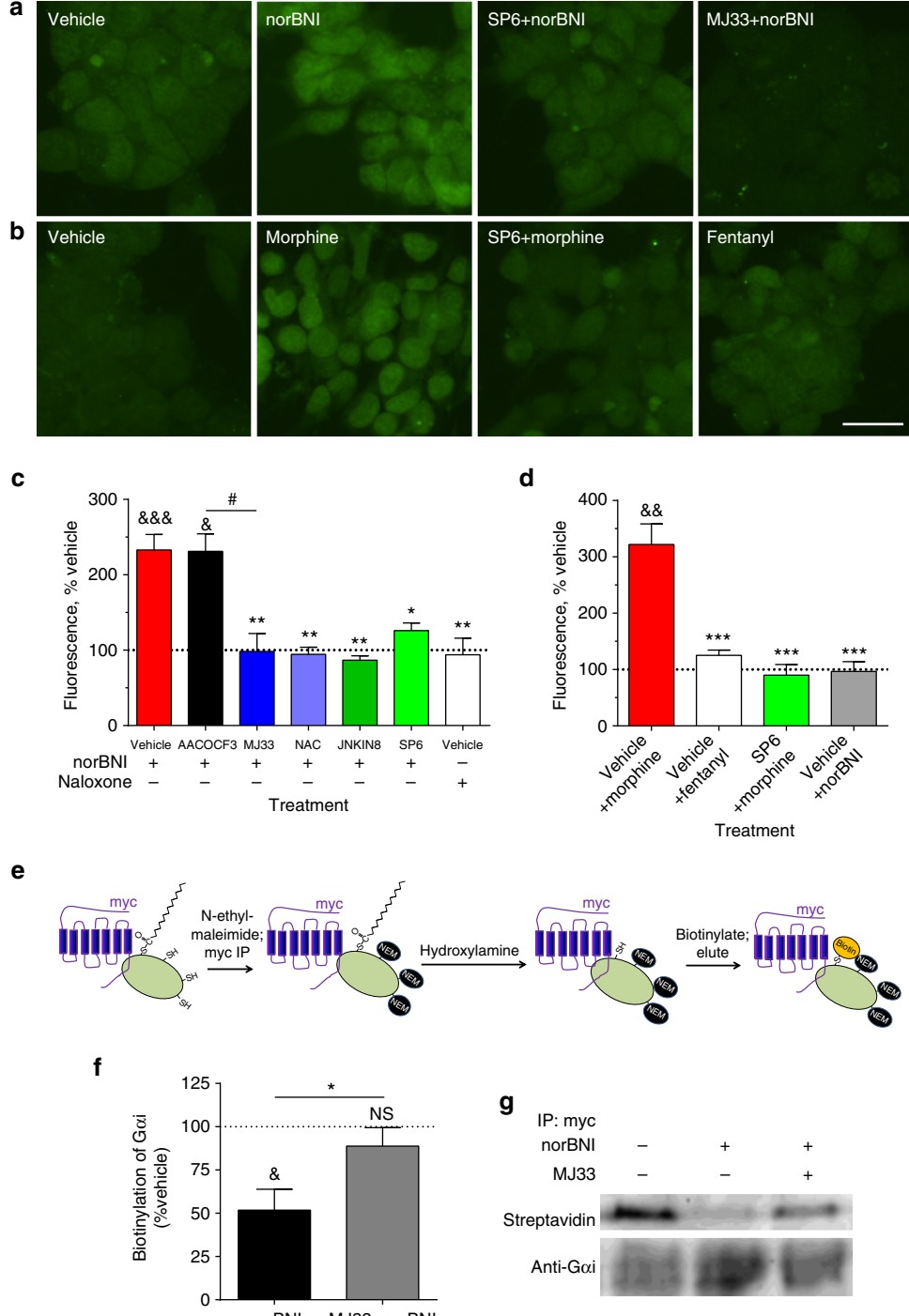

**Fig. 3** NorBNI and morphine induce ROS production via JNK and PRDX6. MycKOR **a**, **c** or mycMOR **b**, **d** expressing HEK293 cells were pretreated with vehicle, AACOCF3 (10 μM), JNK-IN-8 (1 μM), SP610025 (1 μM), MJ33 (10 μM), or N-acetyl-cysteine (NAC, 10 μM) 30 min prior to being treated for 1 h with norBNI (10 μM), morphine (10 μM), fentanyl (10 μM) or naloxone (10 μM). **a**, **b** Cells were imaged for ROS using CellROX Green; *scale bar* represents 12.5 μm. **c** Quantification of ROS in mycKOR cells. NorBNI, but not naloxone, increased CellROX Green fluorescence; this was blocked by pretreatment with MJ33, JNK-IN-8, SP610025, or NAC, but not AACOCF3 (one-way ANOVA, $P < 0.0001$, $n = 3$-11; *,#$P < 0.05$, **$P < 0.01$, Holm–Sidak post hoc analysis). **d** Quantification of ROS in mycMOR cells. Morphine, but not fentanyl or norBNI, increased CellROX Green fluorescence; this increase was blocked by SP610025 (one-way ANOVA, $P < 0.001$, $n = 4$-5; *$P < 0.05$, **$P < 0.01$, Holm–Sidak post hoc analysis). **e** Schematic illustrating the experimental protocol used to generate the results shown in **f**, **g**, and in Supplementary Fig. 5. HEK293 cells stably co-expressing mycKOR and FLAG-GAI3 were pretreated with vehicle, SP610025, or MJ33 and treated 5.5 h with norBNI prior to harvest. Cell membranes were extracted in the presence of N-ethylmaleimide (50 mM) to covalently bind unmodified cysteines, and immunoprecipitated with anti-myc to isolate mycKOR and mycKOR-associated FLAG-Gαi. Myc immunoprecipitates were treated with hydroxylamine to cleave palmitic acids and then incubated with BMCC-biotin to biotinylate previously palmitoylated cysteines. Palmitoylation was measured by probing with streptavidin and normalizing to anti-Gαi immunoreactivity. **f** NorBNI significantly reduced palmitoylation of Gαi coimmunoprecipitated with mycKOR, but not Gαi that was not KOR-associated (&$P < 0.05$, one sample *t*-test with correction for multiple comparisons; $n = 5$-6); this change in palmitoylation was blocked by MJ33 pretreatment (*$P < 0.05$ Student's *t*-test with Welch's correction). **g** Representative immunoblots from **f**. *Error bars* represent mean ± SEM

mice were pretreated with saline or MJ33 and challenged twice with morphine. MJ33 had no effect on the initial antinociceptive response to morphine (Fig. 4c). However, acute tolerance observed as a reduction in antinociceptive response to a second injection of morphine was blocked by pretreatment with MJ33 but not AACOCF3 (10 mg kg$^{-1}$, Fig. 4c, Supplementary Fig. 7b). In contrast, MJ33 had no effect on acute analgesic tolerance to fentanyl (Fig. 4d, Supplementary Fig. 7c). To determine the role of PRDX6 in tolerance to chronic morphine, mice were pretreated twice a day for 5 days with saline or MJ33 2 h prior to morphine. In saline pretreated mice, morphine tolerance was observed by the second injection and continued for 4 days. In contrast, MJ33 significantly reduced tolerance the first 4 days, with no significant reduction in analgesia response to morphine until the third day (Fig. 4e). These findings indicate that JNK-dependent, GRK-independent desensitization of MOR requires PRDX6 activation in vivo, similarly to JNK-dependent inactivation of KOR.

**D2DR stimulates PRDX6-mediated ROS production and tolerance.** To determine if this pathway was a more general mechanism of GPCR regulation, we tested if the Gαi linked D2 dopamine receptor (D2DR) agonists could stimulate ROS production. D2DR and KOR are both implicated in the mesolimbic reward pathway and D2 and KOR are colocalized in ventral tegmental area-accumbal dopamine terminals[30, 31]. The D2DR receptor exists in short (D2DR(S)) and long (D2DR(L)) splice variants that differ in their 3rd intracellular loop; these forms have been suggested to differ in signaling, desensitization and function[32]. The D2 agonist quinpirole significantly increased CellROX Green staining in HA-D2DR(L) (2.7 ± 0.4-fold) expressing HEK293 cells (Fig. 5a, b, Supplementary Fig. 8a, b). The increased staining was also observed in D2DR(S) expressing cells (Supplementary Fig. 8a, b) and blocked by pretreatment with MJ33 (Fig. 5a, b), indicating that the ROS production by quinpirole was PRDX6-dependent. We next asked if PRDX6-mediated ROS production might mediate D2 receptor tolerance in vivo by measuring quinpirole-induced inhibition of locomotor activity. Mice pretreated with quinpirole 2 h prior had significantly greater locomotor activity in the 20 min following a second quinpirole injection than mice pretreated with saline (Fig. 5c, d). This acute tolerance to the hypolocomotor effects of quinpirole was blocked by pretreatment with MJ33. From this, we conclude that PRDX6-stimulated ROS and PRDX6-dependent tolerance may be a more general mechanism of Gαi-GPCR regulation.

**PRDX6 activation mediates GPCR cross-inhibition.** Finally, we hypothesized that local generation of ROS by PRDX6-stimulated NADPH oxidase might cause heterologous desensitization by KOR of nearby GPCRs acting through Gαi. We first tested this in HEK293 cells co-expressing mycKOR and HA-D2DR(L) by measuring ERK1/2 MAPK phosphorylation stimulated by quinpirole[33]. KOR activation by prior norBNI treatment (10 μM, 3–5 h) significantly inhibited quinpirole-stimulated ERK1/2 phosphorylation (27 ± 8.1% of quinpirole stimulation in absence of pretreatment (Fig. 6a, b)). D2DR cross-inhibition was blocked by MJ33 treatment 30 min prior to norBNI, and was not observed in cells only expressing HA-D2DR(L). Cross-inhibition was also observed for mycKOR cells co-expressing D2DR(S) (Supplementary Fig. 8c, d). NorBNI inhibition of quinpirole-stimulated ERK1/2 was not a result of quinpirole acting at KOR, as quinpirole did not stimulate ERK1/2 phosphorylation in cells expressing mycKOR alone (Supplementary Fig. 8e). Together, these results demonstrate that PRDX6 activation by KOR can result in cross-inactivation of D2DR. This heterologous receptor inactivation of D2DR by KOR may have physiological

significance when endogenous ligands such as dynorphin activate JNK.

To determine if norBNI could also inhibit D2DR function in vivo, slice voltammetry was used to measure quinpirole inhibition of stimulated dopamine release in nucleus accumbens[34]. Mice were injected with norBNI (10 mg kg$^{-1}$) or vehicle 1 h prior to harvesting brain slices, and electrically evoked dopamine release was measured within 5–7 h following drug administration. NorBNI pretreatment resulted in a significant, 2-fold rightward-shift in the quinpirole concentration response curve, with an EC$_{50}$ of 53 nM (95% confidence interval 46–61 nM), relative to a vehicle-treated EC$_{50}$ for quinpirole of 28 nM (95% confidence interval 21–34 nM) (Fig. 6c). This shift in potency was prevented by pretreatment with MJ33 (EC$_{50}$ of 40 nM, 95% confidence interval 34–45 nM). These data indicate that KOR activation of JNK/PRDX6 also results in cross-inhibition of D2DR in vivo.

**Discussion**
Our findings demonstrate that JNK activation inhibits the KOR, MOR and D2 receptors by acting on the receptor signaling complexes, which contain proteins defined for KOR by SILAC proteomics and confirmed by immunoprecipitation. JNK activation by norBNI and morphine increased association of the receptor with Gαi and Gβγ, despite preventing nucleotide exchange. Rather than JNK phosphorylating the receptor or G proteins directly, SILAC proteomics revealed that JNK activation specifically increased PRDX6 association with Gαi. PRDX6's PLA2 activity was found to be required for the increased receptor/Gαi association and inhibited receptor function. An important implication of this work is that PRDX6-PLA2 inhibitors or antioxidants may be useful clinically as adjuncts to opioids to reduce the development of tolerance. The proteomic studies also revealed numerous KOR-interacting proteins, including proteins involved in signaling, receptor processing, and degradation; these proteins may also provide insights for future work characterizing the mechanism by which norBNI and similar ligands activate JNK, which remains to be elucidated.

While peroxiredoxins are canonically antioxidant enzymes, PRDX6 also has PLA2 activity that is regulated independently of its antioxidant activity, and inhibition of PRDX6-PLA2 activity may have fewer side effects than targeting both activities[13]. Our studies demonstrate the opioid ligands norBNI and morphine increase ROS in a JNK- and PRDX6-dependent manner. While Gq-coupled receptor activation of PRDX6 has been reported, a role for PRDX6 in opioid signaling had not been previously suggested. Phosphorylation of PRDX6 by p38 and ERK1/2 MAPKs, resulting in PLA2 activity and downstream activation of the NADPH oxidase complex 2 to generate ROS, has been previously described[16]. JNK-dependent regulation of PRDX6 may be through a distinct mechanism, as we were unable to detect PRDX6 phosphorylation, although this may have resulted from an insufficient fraction of phosphorylated PRDX6 for detection by mass spectrometry. NOX2 null mice (which lack a critical subunit of NADPH oxidase 2)[35] and D140A-PRDX6 knock-in mice (which lack PRDX6-PLA2 activity)[36] will be important tools in further studies. It is unclear from the current study if PRDX6 directly interacts with Gαi, but the possibility that JNK recruits PRDX6 as a stable component of the KOR-G-protein complex is intriguing.

This regulatory pathway is not exclusive to opioid receptors, as PRDX6-dependent ROS production and tolerance was also observed for D2DR, and PRDX6-dependent ROS production has been reported for angiotensin receptors[14]. GPCR desensitization by this mechanism appears to be ligand-specific however, as

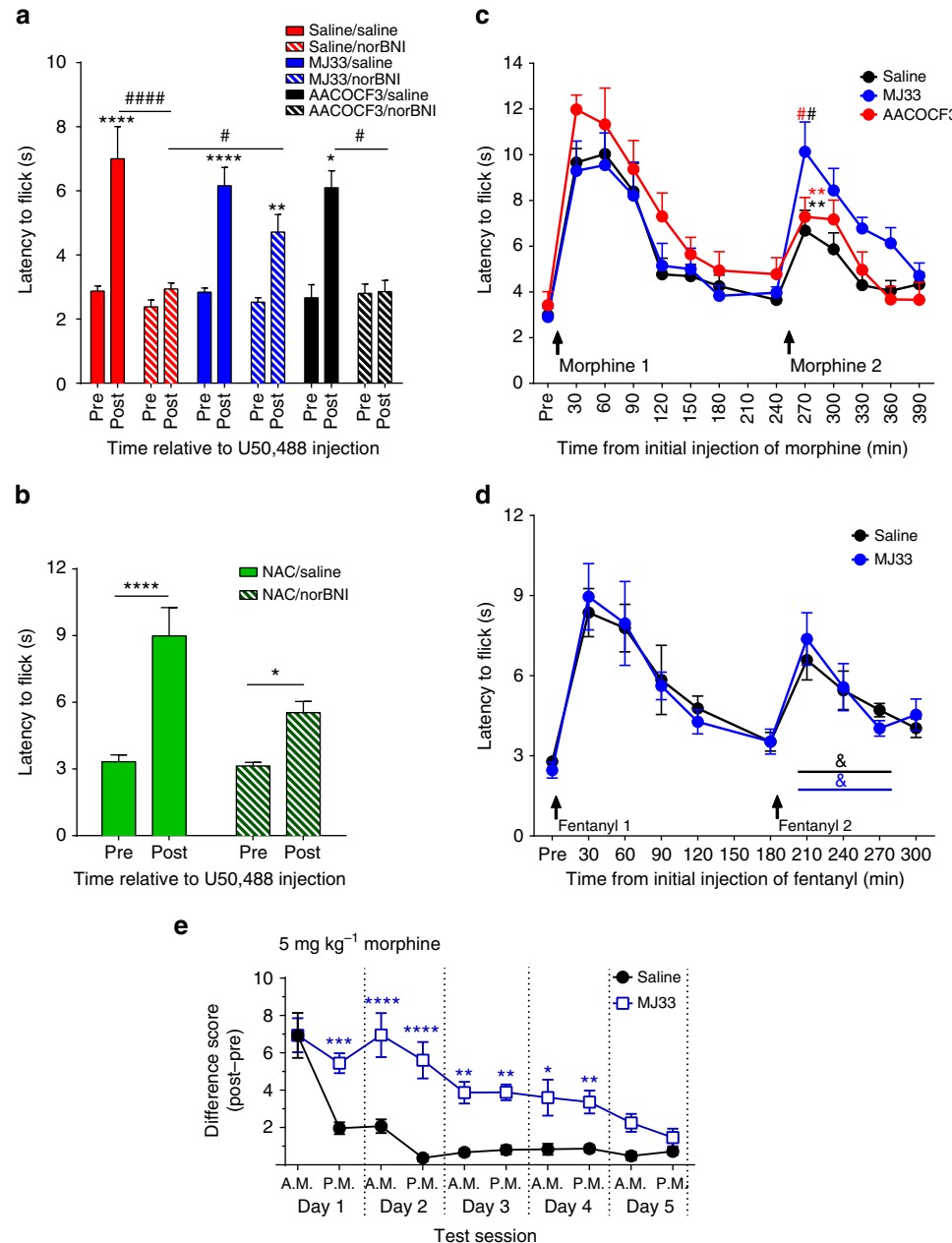

**Fig. 4** PRDX6 mediates the long-lasting effects of norBNI and acute morphine tolerance. **a, b** Mice were injected with saline, MJ33 (1.25 mg kg$^{-1}$), AACOCF3 (10 mg kg$^{-1}$), or NAC (100 mg kg$^{-1}$) 1 h prior to norBNI (10 mg kg$^{-1}$). Three days later, tail withdrawal latency from 52.5 ± 0.3 °C water bath was measured prior to and 30 min after U50,488 (10 mg kg$^{-1}$). **a** The U50,488 stimulated increase in latency was blocked by norBNI following saline or AACOCF3 pretreatment, but not MJ33 (paired two-way ANOVA; significant effect of norBNI, U50,488, and interaction ($P < 0.01$); *$P < 0.01$, **$P < 0.01$, ****$P < 0.0001$, Holm–Sidak post hoc analysis of post vs. pre; #$P < 0.05$ and comparing post-U50,488 latency between groups #$P < 0.05$, ##$P < 0.01$, ###$P < 0.001$, $n = 4$–16). **b** NorBNI failed to block U50,488-stimulated analgesia when mice were pretreated with NAC (paired two-way ANOVA, significant effect of U50,488, norBNI, and interaction ($P < 0.05$; $n = 8$); *$P < 0.05$, ****$P < 0.0001$ Holm–Sidak post hoc). **c, d** Mice were injected with saline, AACOCF3, or MJ33 2 h prior to the initial injection of morphine (10 mg kg$^{-1}$) or fentanyl (0.3 mg kg$^{-1}$). Tail withdrawal latency was measured prior to and at 30 min intervals following morphine or fentanyl. For area under the curve (AUC) values see Supplementary Fig. 7b, c. **c** MJ33 pretreatment increased the analgesic response to the second morphine injection, compared to saline or AACOCF3 (paired two-way ANOVA; significant effect of time, subject matching, and interaction ($P < 0.05$); *(vs. saline (*red*) or AACOCF3 (*black*), $P < 0.05$, # (vs. 30 min, $P < 0.01$, Holm–Sidak post hoc; $n = 6$–8). **d** MJ33 pretreatment had no effect on the analgesic response to a second fentanyl injection (paired two-way ANOVA on second injection; significant effect of time only ($P < 0.0001$), $n = 6$–7; &, significant difference in AUC). **e** Mice were pretreated twice daily for 5 days with saline or MJ33 2 h prior to morphine (5 mg kg$^{-1}$), and tail withdrawal latency was measured prior to or 30 min after morphine. Data are represented as increase in latency following morphine. MJ33 pretreatment increased the analgesic response to repeated morphine and delayed tolerance (paired two-way ANOVA, significant effect of MJ33, morphine treatment, and interaction ($P < 0.0001$); vs. saline *($P < 0.05$), **$P < 0.01$, ***$P < 0.001$, ****$P < 0.0001$, vs. first injection ###$P < 0.001$, ####$P < 0.00001$, Holm–Sidak post hoc; $n = 11$–14). *Error bars* represent mean ± SEM

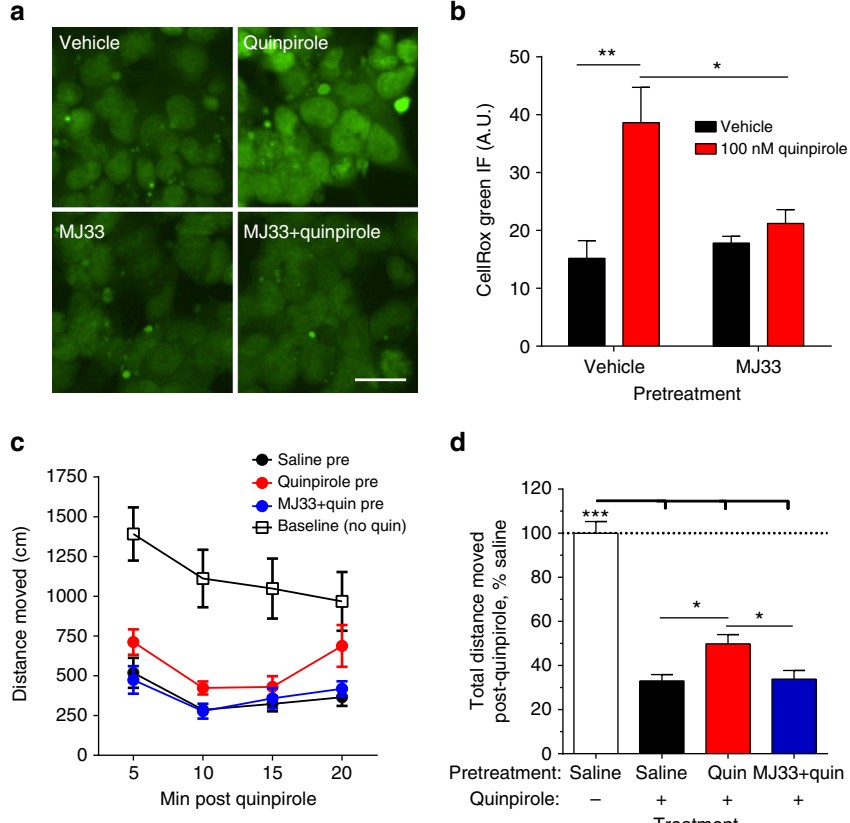

**Fig. 5** D2 dopamine receptor tolerance is mediated by PRDX6-dependent ROS production. **a** HEK293 cells transiently expressing HA-D2DR(L) were pretreated with vehicle or MJ33 (10 μM) 30 min prior to being treated for 1 h with vehicle or quinpirole (100 nM). Cells were imaged for ROS using CellROX Green; *scale bar* represents 12.5 μm. **b** Quantification of ROS in **a**. Quinpirole increased CellROX Green fluorescence; this was blocked by pretreatment with MJ33 (two-way ANOVA, significant effect of quinpirole ($P < 0.01$) and significant interaction ($P < 0.05$), $n = 4–5$; *$P < 0.05$, **$P < 0.01$, Holm–Sidak post hoc analysis of MJ33 vs. vehicle pretreatment). **c**, **d** Mice were injected with saline or MJ33 (1.25 mg kg$^{-1}$ i.p.) 1 h prior to the initial injection of quinpirole (0.2 mg kg$^{-1}$ i.p.) or saline. Two hr later, mice were injected with a quinpirole (0.2 mg kg$^{-1}$ i.p.) or saline (baseline) placed in a novel cage, and locomotor activity was measured over 20 min. **c** Locomotion was measured as distance traveled (cm), in 5 min bins. Pretreatment significantly affected locomotion. (two-way ANOVA, significant effect of treatment ($P < 0.001$), time ($P < 0.001$), and significant interaction ($P < 0.01$), $n = 7–9$). **d** Total locomotion over 20 min following quinpirole administration was quantified as a percent locomotion observed in saline treated mice. Quinpirole significantly reduced locomotor activity in all pretreatment groups, but mice pretreated with quinpirole moved significantly more than mice pretreated with saline or MJ33 + quinpirole (one-way ANOVA, $P < 0.01$; *$P < 0.05$, **$P < 0.01$, ***$P < 0.01$, Holm–Sidak post hoc analysis). *Error bars* represent mean ± SEM

demonstrated by the lack of ROS production and PRDX6-dependent tolerance following fentanyl or naloxone treatment. We hypothesize that arrestin recruitment by fentanyl-activated MOR sterically inhibits Gαi association and prevents PRDX6 induced oxidation, whereas morphine has lower efficacy for arrestin recruitment. This is consistent with prior studies showing the JNK inhibition does not block tolerance resulting from fentanyl, in contrast to morphine and norBNI[4–6, 10]. Further work will be needed to characterize which opioids promote PRDX6/ROS-mediated desensitization, but based on this conception, we predict GPCR ligands that do not efficiently activate GRK/arrestin-dependent receptor desensitization (such as partial agonists or G-protein-biased ligands) may also inactivate Gα signaling through a PRDX6 mechanism. Although our studies initially used transfected HEK293 cells to define this regulatory pathway, the in vivo tolerance studies described establish its physiological relevance.

Unlike prenylation and myristoylation lipoprotein bonding, palmitoylation is a dynamic and biologically reversible modification[25]. Cysteine residues, such as the Cys3 that is palmitoylated in Gαi, are particularly vulnerable to oxidation and cysteine oxidation products range from readily reversible to biologically irreversible[23, 37]. The observed loss of Gαi palmitoylation may be

a result of PRDX6-dependent oxidation of the Cys3 residue that prevents repalmitoylation. The palmitoylation domain does not directly mediate Gα binding with the receptor; however, depalmitoylation results in decreased stability of the Gαi N-terminal helix association with the membrane[24, 25]. Structural studies show that the N-terminal helix of Gα plays a key role in activation-stimulated GDP release by destabilizing the adjacent β1 strand and the α5 helix adjacent to the C terminus, which interfaces the receptor[38, 39]. These results suggest that agonist binding to receptor fails to induce the conformational change in the depalmitoylated Gαi required to stimulate GDP release. Our findings indicate that JNK activation by norBNI-like or morphine-like opioids promotes the PLA2 activity of PRDX6, which in turn increases NADPH oxidase activity and the generation of ROS. This increase in ROS presumably results in oxidation of components of the opioid receptor signaling complex including the observed loss of Gαi palmitoylation. We hypothesize that the cysteine oxidation and depalmitoylation of Gαi results in a tighter association of the heterotrimeric G proteins with the opioid receptors, but a loss of nucleotide exchange function (Fig. 7a, b). Future studies will test this model.

In addition to PRDX6-mediated oxidation and inactivation of G proteins associated with KOR, we demonstrated that activation

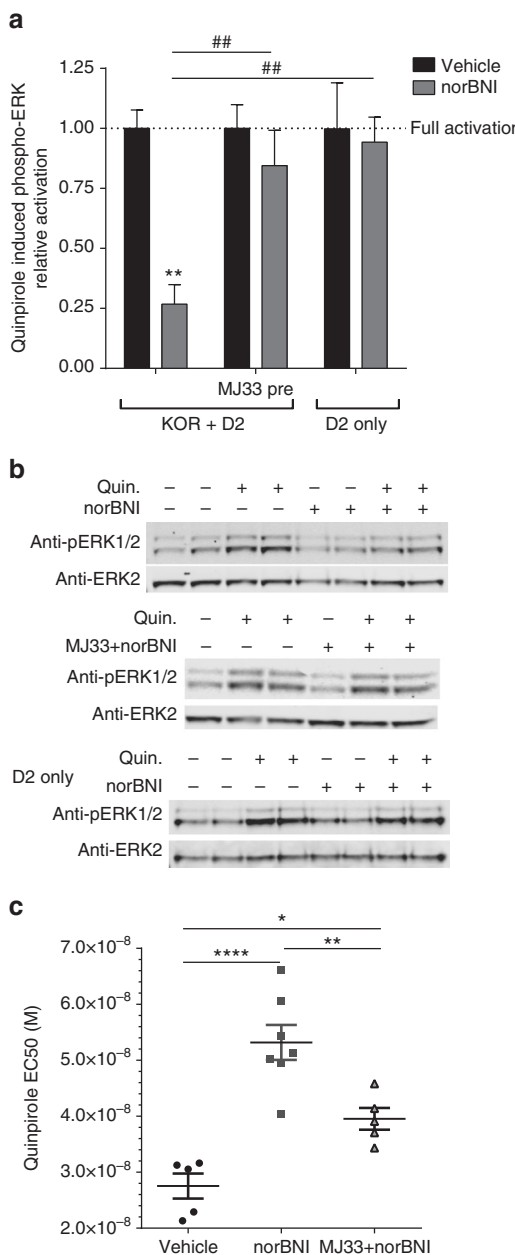

**Fig. 6** PRDX6 mediates GPCR cross-tolerance. **a** MycKOR expressing or wild-type HEK293 cells transiently transfected with HA-D2DR(L) were pretreated with vehicle or MJ33 (10 μM) 30 min prior to being treated for 3-5 h with vehicle or norBNI (10 μM). Cells were then treated with 5 min with vehicle or quinpirole (100 nM) and cell lysates analyzed for phospho-ERK1/2 immunoreactivity. NorBNI reduced quinpirole-stimulated ERK1/2 phosphorylation; the norBNI inhibition of quinpirole response was blocked by MJ33 pretreatment and not observed in cells only expressing HA-D2DR (L). (Two-way ANOVA; significant interaction ($P < 0.05$), significant effect of norBNI ($P < 0.01$), and significant effect of KOR or MJ33 ($P < 0.05$), $n = 5$-8; **$P < 0.01$ Holm–Sidak post hoc analysis of norBNI vs. vehicle pretreatment; ##$P < 0.01$) Holm–Sidak post hoc analysis compared to KOR + D2DR without MJ33 pretreatment.). **b** Representative immunoblots for **a**. **c** EC50 values were calculated from concentration-response curves for quinpirole inhibition of dopamine release 5–7 h after vehicle, norBNI (10 mg kg$^{-1}$, i.p.), or MJ33 (1.25 mg kg$^{-1}$ i.p.) prior to norBNI. NorBNI treatment resulted in a significant increase in quinpirole EC$_{50}$, which was reduced by MJ33 pretreatment (one-way ANOVA, $P < 0.0001$; *$P < 0.05$, **$P < 0.01$, ****$P < 0.0001$, Holm–Sidak post hoc). *Error bars* represent mean ± SEM

of this cascade by norBNI reduced D2DR function both in vitro and in vivo. Based on this finding, GPCR activation of PRDX6 can change neuronal function by silencing of a broad range of GPCRs in a cellular microdomain (Fig. 7c). JNK-dependent inhibition of KOR by the orexin 1 receptor may also be through this mechanism; intriguingly KOR activation had no effect on the Gαq-coupling by orexin receptor[40]. Further studies will be needed to determine whether ROS-mediated GPCR desensitization is specific to Gαi-coupled receptors. ROS have also been shown to regulate several cellular processes, including transcription, cytoskeletal protein polymerization, and tyrosine phosphorylation[41, 42], suggesting ROS may be an important second messenger in GPCR signaling. Furthermore, GPCR-stimulated ROS and ROS-mediated receptor inactivation may play a role in neurodegenerative and psychiatric diseases, which oxidative stress has been implicated[43, 44]. Therefore, this represents an important mechanism of GPCR regulation.

The current work identifies an endogenous regulatory pathway for GPCRs, PRDX6-mediated depalmitoylation of the receptor Gαi subunit, which is dependent upon JNK signaling. This mechanism causes receptor inactivation, and is responsible for the enduring effects of some KOR opioids, such as norBNI, and acute tolerance to some opioid agonists, including morphine. Therefore, beyond the advancement of our knowledge of basic pharmacological mechanisms, the identification of this mechanism has important direct clinical implications. Resolving the mechanism of the enigmatically long-lasting antagonist effects of compounds such as norBNI removes one of the barriers for advancing these compounds into the clinic. Identifying one of the

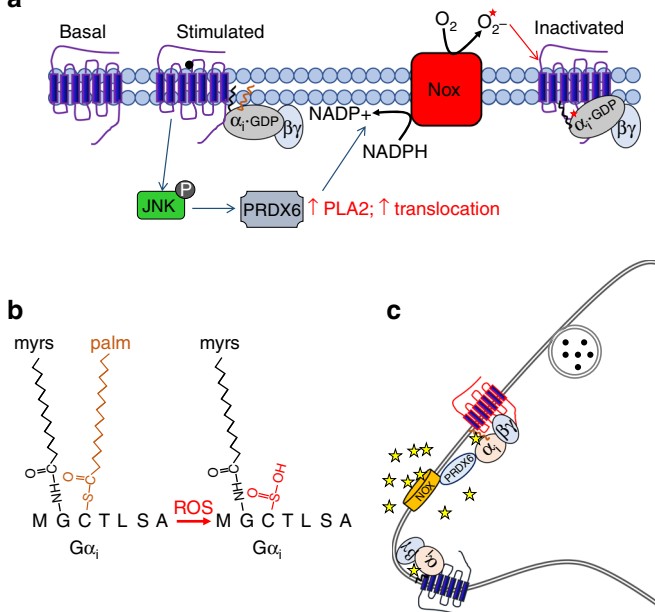

**Fig. 7** Model of GPCR inactivation by PRDX6-mediated oxidation and depalmitoylation. **a** Proposed model of JNK-mediated Gαi regulation by norBNI- and morphine-like drugs. JNK promotes membrane localization of PRDX6 and PLA2 activity, thereby locally activating NADPH oxidase (NOX) and stimulating the generation of ROS. The Gαi-PRDX6 interaction may help promote PRDX6 translocation and/or locally target PLA2 activity. Gαi oxidation disrupts Gαi palmitoylation, jamming the G-protein receptor complex in an inactive conformation. **b** Oxidation of cys-3 of Gαi blocks repalmitoylation of the residue. **c** Proposed model of GPCR cross-regulation by this mechanism. ROS generated as result JNK/PRDX6 activation by GPCR (i.e., KOR) diffuse locally, oxidizing nearby GPCR-Gαi complexes, function as paracrine or intracrine signals to silence inhibitory GPCRs

mechanisms responsible for acute morphine tolerance and demonstrating that it can be blocked by a small-molecule inhibitor of PRDX6, opens up the possibility for adjunct treatment to reduce acute tolerance to morphine.

## Methods

**Animals**. Male C57BL/6 mice (adult, 20–30 g) (Charles River Laboratories) were group housed (2–3 per cage) and maintained on a 12 h light/dark cycle with food and water ad libitum. All animals were drug naive with no prior procedures performed. Animal procedures were approved by the Animal Care and Use Committee of the University of Washington and conform to the guidelines on the care and use of animals promulgated by the National Institutes of Health.

**Cell culture**. HEK293 cells (American Type Culture Collections) stably expressing KORGFP, mycKOR, mycMOR, HA-D2DR, mycKOR and HA-D2DR, mycKOR and HA-PRDX6, or mycKOR (puro) and FLAG-Gαi3 were generated by transfecting HEK293 cells using Fugene HD (Promega) according to manufacturer instructions. HEK293 cells were maintained in Dulbecco's modified medium/F12 (Fisher Scientific) with 10% fetal bovine serum (Sigma-Aldrich) and penicillin-streptomycin-L-glutamine (Fisher Scientific). HEK293 cells expressing mycKOR or mycMOR were grown in media supplemented with G418 (200 μg ml$^{-1}$) (Fisher Scientific); HEK293 cells expressing HA-D2DR were grown in media supplemented with hygromycin (50 μg ml$^{-1}$) (Fisher Scientific); HEK293 cells expressing mycKOR and HA-D2DR or HA-PRDX6 were grown in media supplemented with G418 (200 μg ml$^{-1}$) and hygromycin (50 μg ml$^{-1}$); HEK293 cells expressing mycKOR (puro) and FLAG-GNAI3 were grown in media supplemented with puromycin (0.5 μg ml$^{-1}$) (Sigma-Aldrich) and G418 (200 μg ml$^{-1}$). For SILAC cell culture, cells were grown for five doublings in 15 cm plates in custom DMEM media (Caisson Labs) containing isotope-labeled lysine and arginine (light media or L:R0K0 = lys0/arg0; medium media or M:R6K4 = lys4/arg6; heavy media or H: K10R8 = lys8/arg10) (Cambridge Isotope Labs).

**Drugs**. DAMGO ([D-Ala2, N-MePhe4, Gly-ol]-enkephalin) (Sigma-Aldrich) and U69,593 (NIDA Drug Supply) were dissolved in GTPγS binding buffer (50 mM HEPES, 100 mM NaCl, 5 mM MgCl$_2$, 1 mM EDTA, 0.1% BSA, 1 mM DTT, pH 7.4). For cell culture studies, (−)-quinpirole hydrochloride (Sigma-Aldrich), (−)-U50,488 (Tocris), morphine sulfate (NIDA Drug Supply), fentanyl (NIDA Drug Supply), norBNI (NIDA Drug Supply), MJ33 (Sigma-Aldrich), N-acetyl-L-cysteine (Millipore), and naloxone (NIDA Drug Supply) were dissolved in H$_2$O. AACOCF3 (Tocris) was purchased in an ethanol solution and diluted in H$_2$O. SP600125 (Millipore) and JNK-IN-8 (Fisher Scientific) were dissolved in DMSO. For in vivo studies, (−)-quinpirole hydrochloride, morphine sulfate, fentanyl, ± U50,488 (NIDA Drug Supply), norBNI, MJ33, AACOCF3, and N-acetyl-L-cysteine were dissolved in 0.9% saline.

**Constructs**. MycKOR and mycMOR were made by inserting rat KOR or MOR sequences into pcDNA3.1-myc (provided by Dr Nephi Stella, University of Washington). The rat KOR cDNA sequence was amplified by PCR adding 5′ *Bcl*I and 3′ *Not*I restriction sites and then cut at *Bcl*I and *Not*I. The rat MOR cDNA sequence was amplified by PCR adding 5′ *Bcl*I and 3′ *Not*I restriction sites and then cut at *Bgl*II and *Not*I. KOR or MOR were subcloned into pcDNA3.1-myc at the *Bam*HI and *Not*1 sites. MycKOR-puro was synthesized by amplifying mycKOR by PCR, adding 5′ *Nhe*I and 3′ *Sac*I sites and subcloning into pIRES-EGFP-puro (Addgene plasmid #45567). Plasmid GNAI300000, expressing Gαi3 under the CMV promoter, was purchased from the University of Missouri-Rolla cDNA Resource Center[45]; FLAG-Gαi3 was made by inserting a spacer-flanked FLAG sequence (SGGGGSDYKDDDDKSGGGGS) between amino acid 91 and 92, a site where tag insertions have been previously shown not to disrupt protein function[46, 47]. The insertion was made via Gibson assembly according to manufacturer instructions (New England Biolabs). HA-D2DR(L) plasmid was purchased from Sino Biological. HA-D2DR(S) was made by amplifying the D2s coding region from a pcDNA-D2s-L-Venus (Addgene Plasmid #19966) with primers that incorporated 5′ *Hind*III and 3′ *Not*I restriction sites and an HA tag. The HA-D2DR (L) coding sequence was excised using *Hind*III and *Not*I restriction sites and the amplified HA-D2DR(S) was subcloned into the cassette at the *Hind*III and *Not*I sites. PRDX6 tagged with HA at the C-terminus was purchased from Sino Biological. The pC1-HyPer-Red plasmid was purchased from Addgene (Addgene #48249)[21].

**Spinal cord membrane preparation**. The lumbar region of the spinal cords of wild-type mice were removed and homogenized in 2 ml membrane buffer (300 mM NaCl, 1 mM EDTA, 1 mM Na$_3$VO$_4$. 1 mM NaF, 1x protease inhibitor (Millipore), 1x phosphatase inhibitor (Fisher Scientific) using a Polytron homogenizer at setting 1 (Kinematica). Homogenates were centrifuged at $15,000 \times g$ at 4 °C for 20 min. Supernatant was discarded and the pellet was washed in 2 ml membrane buffer, rehomogenized, and recentrifuged twice before freezing the pellet at −80 °C until use. Membrane pellets were resuspended in 550 μl GTPγS binding buffer and protein concentration was determined on the day of experimentation.

**Kinase reaction**. JNK1α1 (Millipore) and GST-ATF-2 (Millipore) were diluted in kinase dilution buffer (50 mM Tris/HCl, 0.1 mM EGTA, 0.1% 2-mercaptoethanol, 0.1% bovine serum albumin (BSA), pH 7.5) 10 μg membrane proteins or 2.7 μg GST-ATF2 (used as a positive control for JNK enzymatic activity) were incubated with 160 ng active JNK1α1 or an equivalent volume of kinase dilution buffer in 20 μl kinase reaction buffer (50 mM Tris/HCl, 150 mM NaCl, 50 mM MgCl, [+ or −] 0.5 mM ATP, 0.1 mM EGTA, 0.5 mM DTT, 1 mM Na$_3$VO$_4$, pH 7.5) at 30 °C for 30 min. The reaction was terminated by placing tubes rapidly on ice and samples were immediately assayed for [$^{35}$S]GTPγS binding.

**GTP binding measurements**. [$^{35}$S]GTPγS (Perkin-Elmer), GDP (Sigma-Aldrich), and GTPγS (Fisher Scientific) were dissolved in GTPγS binding buffer (50 mM HEPES, 100 mM NaCl, 5 mM MgCl$_2$, 1 mM EDTA, 0.1% BSA, 1 mM DTT, pH 7.4). 10–20 μg membrane proteins from in vitro JNK kinase assay treated preparations were incubated with 1 μM DAMGO, 1 μM U69,593, 1 μM GTPγS, or an equivalent volume of GTPγS binding buffer in 50 mM GTPγS binding buffer) at 30 °C for 1 h in the presence of 0.1 nM [$^{35}$S]GTPγS and 10 mM GDP. Bound [$^{35}$S] GTPγS was separated from free [$^{35}$S]GTPγS by rapid filtration using a Brandel cell harvester onto GF/B filters (Brandel) pre-soaked in 50 mM Tris/0.1%BSA. Filters were washed 3x using 50 mM Tris/0.1%BSA. Bound [$^{35}$S]GTPγS was measured using a liquid scintillation counter. Data were normalized to percent of binding in vehicle-treated controls with cold GTPγS CPMs subtracted from stimulated CPMs.

**Co-immunoprecipitation**. HEK293 cells, stably expressing the indicated constructs, were grown in SILAC labeled media[11] containing dialyzed FBS (Sigma-Aldrich), starved in serum-free SILAC media, and treated 5.5 h with vehicle or norBNI (10 μM). Cells were scraped in cold PBS, pelleted, and solubilized in 1% digitonin (Sigma-Aldrich) in coIP buffer (50 mM Tris, pH 7.5, 150 mM NaCl, 5 mM EDTA, 1 mM sodium orthovanadate, 1 mM NaF, protease and phosphatase inhibitor cocktail). Twelve micrograms of protein were immunoprecipitated overnight at 4 °C with 150 μl anti-myc or anti-FLAG agarose slurry (Sigma-Aldrich), washed, eluted with excess myc or 3xFLAG peptide (Sigma-Aldrich), and analyzed by mass spectrometry as described below; specific interacting proteins were identified as proteins significantly enriched as compared to a specificity control in which excess myc or FLAG peptide was added to the immunoprecipitation. Co-immunoprecipitation samples for western analysis were prepared similarly, but cells were grown in normal media and pretreated as indicated. Cells were pretreated with SP610025 (1 μM) or DMSO, or MJ33 (10 μM) or vehicle, 30 min prior to treatment with norBNI (10 μM), naloxone (10 μM) or vehicle.

**SILAC-based mass spectrometry**. Individual SILAC samples for the forward experiment and for the reverse experiment were combined during elution at a ratio of 1:1:1, and concentrated using a 10 kD spin column (Millipore). Samples were treated with 1 mM tris (2-carboxyethyl)phosphine and 2 mM choloroacetamide to reduce proteins and cap cysteines respectively, diluted with LDS sample buffer (Life Technologies) supplemented with 2% 2-mercaptoethanol, and run on 4–12% Bis-Tris precast gels (Life Technologies) at 75 V. Proteins were visualized with coomassie blue staining, and gel lanes were cut into five slices prior to overnight digestion with MS-grade trypsin (Fisher Scientific). Peptides were then extracted and desalted on C18 stagetips (3 M) and stored at 4 °C until processing. Peptides were separated on 3 μm reverse phase C18 material (Reprosil C18.aq, Dr Maisch) packed in a 75 μm i.d. 15 cm long column with a 10 μm pulled tip using 90 min gradients of 3–35% acetonitrile at 200 nl/min. LC solvent A was 0.1% acetic acid and solvent was 0.1% acetic acid, 99.9% acetonitrile. LC-MS data was collected with a Thermo Dionex RSLCnano pump and a Thermo Orbitrap Elite (Fisher Scientific). MaxQuant v1.3.05[12] and a Uniprot human database were used to identify and quantify proteins, using protein, peptide, and site FDRs of 0.01. MS database search parameters were: trypsin/P enzyme with up to 2 missed cleavages, oxidized methionines as a variable mass modification, and carbamidomethylated cysteines as a fixed mass modification. Data were further analyzed using the Perseus software package (version 1.4.1.3[12]). For the forward and reverse experiments, significance B (taking into account normalized ratios and summed protein intensity) was calculated for log2 H/L, H/M, and M/L values.

**Western analysis of co-immunoprecipitation**. Eluates were loaded on 4–12% Bis–Tris precast gels and run at 80–100 V for 3 h. Blots were transferred to nitrocellulose (Whatman) for 1.5 h at 30 V. Nitrocellulose was blocked with 5% BSA-TBST (20 mM Tris, 150 mM NaCl, 0.1% tween-20, pH 7.2) 1 h at room temperature and stained overnight for myc and Gαi (myc IP), or FLAG and KOR (KT2[48]) (FLAG IP). Blots were incubated in IRdye secondary antibody (Li-Cor Biosciences, 1:10,000) in 1:1 Odyssey buffer (Li-Cor Inc) and 5% milk-TBST 1 h at room temperature and then scanned on the Odyssey Infrared Imaging System (Li-Cor, Inc.). Band intensity was measured using the Odyssey software (Image Studio 3.1 and Image Studio Lite 4.0, Li-Cor, Inc.) and expressed as Gαi band intensity over myc band intensity (myc IP) or KOR band intensity over FLAG band intensity (FLAG IP). Data were normalized to percentage of control sample (% vehicle) and plotted using GraphPad Prism 6.07 (GraphPad Software, Inc.). Statistical significance ($P < 0.05$) was determined by Student's $t$-test or analysis of variance followed by Holm-Sidak post hoc test.

**Western analysis of protein phosphorylation.** HEK293 cells stably expressing the indicated constructs were serum starved 6 h prior to drug treatment. Cells were treated as described in figure legends and then lysed in lysis buffer (50 mM Tris-HCl pH 7.5, 300 mM NaCl, 1 mM EDTA, 1 mM $Na_3VO_4$, 1 mM NaF, 10% glycerol, phosphatase and protease inhibitors). Lysates were sonicated and centrifuged ($15,000 \times g$, 20 min, 4 °C), and the supernatant was stored at –20 °C. Total protein concentration was determined by BCA assay (Pierce) with bovine serum albumin standards before loading 40 µg (JNK) 30 µg (cJun), or 20 µg (ERK1/2) onto 10% Bis–Tris precast gels (Life Technologies) and running at 130 V for 1.5–2 h. Protein was transferred to nitrocellulose and immunoblotted. Phospho-protein band intensity was normalized to total (ERK1/2, cJun) or actin (JNK) band intensity, and expressed as a % vehicle.

**Western analysis of quinpirole-stimulated phospho-ERK.** HEK293 cells stably expressing mycKOR were transiently transfected with HA-D2DR(L) 24–48 prior to the experiment using Fugene HD. Cells serum starved 4 h prior and then treated with MJ33 (10 µM) or vehicle 30 min prior to 3–5 h treatment with norBNI (10 µM). Cells were then treated with quinpirole (100 nM) 5 min before lysis. Cells were lysed and proteins processed for western analysis as described in supplemental methods. Blots were probed with phospho-ERK1/2 and ERK1/2 primary antibodies. Band intensity quantified as phospho-ERK1/2 band intensity over ERK1/2 band intensity, and the percent increase stimulated by quinpirole was expressed as a ratio of the increase with no pretreatment.

**Western antibody conditions.** Myc (Cell Signaling #2276), Gαi (Cell Signaling #5290), phospho-JNK (Cell Signaling #9251), phospho-ERK1/2 (Cell Signaling 9101), phospho-cJun (Cell Signaling 9261), cJun (Cell Signaling 2315), FLAG (Sigma-Aldrich F1804), HA (Sigma-Aldrich H3663), and ERK2 (Santa Cruz sc1647) antibodies were incubated used 1:1000 in 5% BSA-TBST overnight at 4 °C. Actin (Abcam AB8226 or AB8227) antibody was incubated 1:5000 in 5% BSA-TBST. KT2 antibody was incubated 1 µg ml$^{-1}$ in 5% BSA-TBST.

**PLA2 activity.** HEK293 cells stably expressing mycKOR and HA-PRDX6 were washed in PLA2 buffer (50 mM Tris, 100 mM NaCl, 1 mM EGTA, pH 7.0), and then sonicated in PLA2 buffer supplemented with protease and phosphatase inhibitors. For whole cell protein, samples were centrifuged at $500 \times g$ for 3 min to remove unbroken cells, and the supernatant collected. For the cytosolic fraction, samples were centrifuged $15,000 \times g$ for 30 min, and the supernatant collected. This pellet was sonicated in PLA2 buffer supplemented with protease and phosphatase inhibitors, centrifuged at $500 \times g$ for 3 min, and the supernatant collected as the membrane fraction. Protein (from whole cell, membrane, or cytosol fractions) was quantified by BCA assay and 50–100 µg protein (identical across groups within an experiment) was measured for PLA2 activity (pH 7.0, in absence of calcium) using the BODIPY-phosphatidyl choline (BODIPY-PC) based EnzChek Phospholipase A2 kit (Molecular Probes). At 15–20 min, samples were analyzed in a SpectraMax M2e (Molecular Probes) at excitation 460 nm, emission 515 and 575 nm. Background was subtracted and the ratio of 515/575 nm emission analyzed as a percent change from vehicle.

**Immunocytochemistry.** HEK293 stably expressing mycKOR were grown on poly-D-lysine treated coverslips the day prior to the experiment. Cells were serum starved 5 h, then treated with SP610025 (1 µM) or vehicle 30 min prior to 1 h treatment with norBNI (10 µM). Cells were rinsed in PBS and fixed 15 min with 4% PFA prior to immunostaining for myc and PRDX6. Cells were blocked 30 min with 1% bovine serum albumin in PBS, 0.025% triton-X100, incubated in mouse anti-myc antibody (Cell Signaling 2276, 1:500) and rabbit anti-PRDX6 antibody (Abcam ab59543, 1:250) overnight at 4 °C, washed, and incubated with alexafluor 555 anti-mouse and alexafluor 488 anti-rabbit secondary 1 h. Cells were washed, mounted on glass slides with VectaShield HardSet with DAPI, and stored at 4 °C until imaging. All images for a given experiment were captured with the same gain and offset settings on a Leica confocal microscope. Images were imported into ImageJ for processing. Colocalization was analyzed by drawing a line through a cell, obtaining a plot profile in each channel, and calculating the Pearson correlation coefficient for the two immunostains. For each replicate, norBNI and vehicle-treated samples were processed in parallel, and each replicate represents an average value from 7 to 10 cells.

**Western analysis of PRDX6 translocation.** HEK293 cells stably expressing mycKOR and HA-PRDX6 were sonicated in PLA2 buffer supplemented with protease and phosphatase inhibitors. Samples were centrifuged $15,000 \times g$ for 30 min, and the supernatant (cytosol) collected. This pellet was washed in PLA2 buffer, and again centrifuged $15,000 \times g$ for 30 min. This pellet was sonicated in PLA2 buffer supplemented with protease and phosphatase inhibitors, 1% triton-X100, and 0.1% SDS, and then centrifuged at $500 \times g$ for 3 min. The supernatant collected as the membrane fraction. Membrane protein was quantified by BCA assay before loading 25 µg onto 10% Bis-Tris precast gels (Life Technologies) and running at 130 V for 1.5–2 h. Protein was transferred to nitrocellulose and immunoblotted for HA and actin. HA-PRDX6 band intensity was normalized actin band intensity, and expressed as a % vehicle.

**Reactive oxygen species.** HEK293 cells stably expressing mycKOR or mycMOR were grown on poly-D-lysine treated cover slips the day prior to the experiment. For D2DR experiments, HEK293 cells were plated on coverslips 48 h prior to the experiment and transiently transfected with HA-D2DR(L) 24 h prior. Cells were serum starved 5 h, then treated with MJ33 (10 µM), SP610025 (1 µM), naloxone (10 µM), or vehicle 30 min prior to 1 h treatment with norBNI (1 µM or 10 µM). CellROX Green (10 µM, Molecular Probes) was added during the last 30 min of treatment. Cells were rinsed in PBS and fixed 15 min with 4% PFA. Cells were mounted on glass slides with VectaShield HardSet with DAPI (Vector Laboratories), and imaged within 12 h. Cover slips were imaged on a Nikon upright fluorescent microscope with Nikon Elements AR v3.1 software (Nikon Instruments). To prevent photoactivation, exposure to light during sample generation and imaging was minimized. Two representative fields from each cover slip were imaged for CellROX (488 nm) and DAPI. Exposure times were held constant for every fluorophore throughout the entire experiment. Image intensities for between 7 and 20 cells per image were quantified using ImageJ v 1.42q (National Institute of Health), and these were averaged to give an average field intensity value. This was done for the two cover slip images and averaged to make one sample ($n$).

**Reactive oxygen species imaging using HyPerRed.** MycKOR expressing HEK293 cells grown on poly-D-lysine coated coverslips were transiently transfected with HyPerRed using Fugene HD. The following day, cells were serum starved 6 h and treated as described. Cells were blocked 30 min with 1% bovine serum albumin in PBS, 0.025% triton-X100, incubated in mouse anti-KOR antibody overnight at 4 °C, washed, and incubated with Alexafluor 488 anti-rabbit secondary (Life Technologies) 1 h. Cells were washed, mounted on glass slides with VectaShield HardSet with DAPI, and stored at 4 °C until imaging. Cover slips were imaged on a Nikon upright fluorescent microscope. 10x, representative fields from each cover slip were imaged for KOR, HyPerRed (555 nm) and DAPI. Exposure times were held constant for every fluorophore throughout the experiment. Images were adjusted to a standard threshold value, and cell counts were taken using ImageJ.

**PRDX6 siRNA knockdown.** MycKOR expressing HEK293 cells were transiently transfected with control siRNA (Fisher Scientific #4390843) or predesigned validated siRNA against PRDX6 from Fisher Scientific (combination of s18428 and s18429, Fisher Scientific #4427038 and #4390824). Transfections conditions were selected based on obtaining 50% knockdown in an initial screen, as quantified by western blot 48 h post-transfection. Cells at 50% confluency in six-well plates were transfected with 250 pmol siRNA and 5 µl Lipofectamine 2000 (Life Technologies) overnight, then replated on poly-D-lysine coated coverslips. The day after plating on coverslips, cells were treated and imaged the following day as described. Knockdown was verified by staining coverslips with PRDX6 antibody (Abcam #ab59543, 1:250).

**Palmitoylation assay.** Palmitoylation was assayed using the acyl-biotin exchange method[27]. HEK293 cells stably expressing mycKOR and FLAG-Gαi3 were treated 30 min with vehicle or 10 µM MJ33 prior to 5.5 h treatment with 10 µM norBNI or vehicle. Cells were then solubilized and immunoprecipitated with anti-myc agarose overnight as described, except coIP buffer was supplemented with 50 mM N-ethylmaleimide (NEM) (Sigma-Aldrich). Unbound lysate from the myc immunoprecipitation was saved and immunoprecipitated with anti-FLAG agarose. Both myc immunoprecipitates ("KOR bound") and FLAG immunoprecipitates ("unbound") were assayed for palmitoylation using the acyl-biotin exchange assay. Beads were washed and incubated in coIP buffer (pH 7.5, plus 10 mM NEM) 10 min prior to washing 3x with coIP buffer (pH 7.2, no NEM). During the last wash, 20% of beads were transferred to a no-HAM negative control; beads were then incubated 1 h at room temperature in HAM buffer (coIP buffer pH 7.2, plus 0.5 M hydroxylamine (Fisher Scientific)) or coIP buffer (pH 7.2) alone. Beads were washed once in coIP buffer (pH 6.8), then incubated 1 h at 4 °C in biotin buffer (coIP buffer pH 6.8; plus 5 µM biotin-BMCC (Fisher Scientific)). Beads were washed once in coIP buffer (pH 6.8) and then eluted with myc peptide or 4× LDS buffer. Samples were analyzed by western blot, detecting KOR or Gαi as described above and biotin with Streptavidin IRDye800 (Li-Cor, Inc.).

**Analgesia.** Antinociception was measured using the warm water tail-withdrawal assay. Mice were pre-injected intraperitoneal (i.p.) with saline, MJ33, NAC, AACOCF3, or norBNI as described. The response latency for an animal to withdraw its tail after being immersed in 52.5 ± 0.3 °C water was measured before and at 30 min intervals after treatment with U50,488 (10 mg kg$^{-1}$, i.p.), morphine (10 mg kg$^{-1}$, i.p.), or fentanyl (0.3 mg kg$^{-1}$, i.p.). A 15 sec maximal immersion duration was used as a cut off to prevent tissue damage. For chronic morphine, each day mice were pre-injected with MJ33 or saline and injected with morphine 2 h later, twice a day (morphine injections between 10:30 and 11:00 am and between 2:30 and 3:00 pm). Published literature[49] suggests low toxicity of MJ33 over 4 days, and no adverse responses were observed in this study.

**Locomotor activity.** Mice were pretreated with vehicle or MJ33 (1.25 mg kg$^{-1}$ i.p.) 1 h prior to a first quinpirole (0.2 mg kg$^{-1}$ i.p., or vehicle) treatment. After 2 h, all

groups were treated with quinpirole (0.2 mg kg$^{-1}$ i.p.) and immediately placed in individual, novel home cage bottoms for 20 min. Locomotor activity was recorded by video, and analyzed for distance traveled using Ethovision v3.013 (Noldus).

**Fast-scan cyclic voltammetry.** Mice were rapidly decapitated and the head placed in pre-oxygenated, cold modified artificial cerebrospinal fluid (248 mM sucrose in place of NaCl). The brain was removed, and 250 µm coronal slices containing nucleus accumbens were prepared. To measure evoked dopamine, nucleus accumbens slices were transferred to standard oxygenated artificial cerebrospinal fluid and incubated for 1 h at 37 °C prior to holding at room temperature before recordings. Slices were placed in a recording chamber, and perfused with oxygenated artificial cerebral spinal fluid (124 mM NaCl, 2.5 mM KCl, 1.25 mM NaH$_2$PO$_4$, 1.25 mM MgSO$_4$, 2 mM CaCl$_2$, 10 mM dextrose, 25 mM NaHCO$_3$) at 31–33 °C throughout recording. Carbon-fiber working electrodes were fabricated as described[50], hand-cut to 100–150 µm past the capillary tip. A single biphasic electric pulse (100–300 µA, 2 ms per phase) was applied to a parallel bipolar stimulating electrode to evoke dopamine release, and the potential at the working electrode was held at −0.4 V versus Ag/AgCl, ramped to + 1.3 V and back to −0.4 V every 100 ms. Baseline dopamine release was measured every 2 min for 20 min prior to measuring release after increasing concentrations of quinpirole (20 min per concentration). Waveform generation, data acquisition and analysis were carried out on a PC-based system using uses two PCI multifunction data acquisition cards and software written in LabVIEW v7.1 (National Instruments). The last 8 min (four recordings) were used for calculating a quinpirole concentration response curve in GraphPad Prism, using a three parameter least squares nonlinear regression with top constrained to 100%.

**Quantification and statistical analysis.** GraphPad Prism 6.07 software (GraphPad Software, Inc.) was used for data analysis. Data are represented as mean ± SEM. When data is described as % control or % baseline, data are normalized to a within-replicate control set at 100%, which is represented by a dashed line. Statistical analyses are described in the corresponding figure legends. Uncropped western blots for representative western images are shown in Supplementary Figs 9 and 10.

**Data availability.** The authors declare that all the data supporting the findings of this study are included in the article and its supplementary data files and available from the corresponding author upon reasonable request.

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

## Acknowledgements

This work was supported by a USPHS grants PO1-DA35764, T32-DA07278, P30-DA28846, and KO5-DA20570 from the National Institute on Drug Abuse. We thank Daniel Messinger, Allisa Song, Hyong Won Suh, and Martin Golkowski for technical assistance. We thank Haripriya Shankar and Richard Gardner for advice at an early stage of this project. The authors have no significant financial conflicts of interest to declare.

## Author contributions

S.S.S. performed most of the studies and analyzed the data. B.B.L. and K.L.R. did the CellROX and HyPerRed imaging. B.B.L. and A.D.A. did the behavioral assays. L.M.B., K.L.R. and P.E.M.P. did the FSCV studies. J.R.K. did the GTPγS binding assays. S.E.O. did the proteomic analysis. S.S.S. and C.C. conceived the study and wrote the manuscript, and B.B.L. provided important discussion.
