## [Peer Review File · Nature Communications]

Reviewers' comments:

Reviewer #1 (Remarks to the Author):

This submission examines a novel pathway involved with inactivation/inhibition of GPCR responses. It is based upon earlier work from the same laboratory indicating that the prolonged inhibition of kappa actions by antagonists such as norBNI result from activation of JNK. The current submission extends this to provide a mechanism whereby that results in formation of the generation of reactive oxygen species produced by peroxiredoxin6 (PRDX6), which in turn results in the depalmylation of Gai. The work is carefully carried out with the appropriate controls and presents a logical extension of the earlier findings. The authors extend this to show evidence of heterologous desensitization between D2 dopamine receptors and opioid receptors. Technically, the work is well described and the conclusions supported by the evidence.

Some issues that require some discussion:

1. A major question involves why a kappa antagonist and a mu agonist elicit the same response. The actions of norBNI would suggest that the pathway is not initiated by activation of the receptor. How do they envisage this response being initiated? Is it possible that norBNI is interacting with an allosteric site or even an alternative target? It would be helpful to the reader to have a better understanding of the mechanisms involved if they are known.
2. In Fig. 4c, it appears that the peak MJ33 effect is similar, but it appears to diminish more slowly. It would help if the authors determined the area under the curve (AUC) to compare the full effect of the drug rather than just peak effect
3. The authors examine 'acute tolerance'. While this is a well established entity, it is not clear that it has any relationship to the tolerance produced by treating subjects for weeks to months, which is the clinical situation. The authors should discuss this and, if possible, look at the effects of chronic morphine to see if they replicate those seen acutely.
4. The authors point out the difference between fentanyl and morphine. While they allude to possible differences in B-arrestin signaling and partial agonists, this should be explained in greater detail. How do the biased factors of the compounds compare? Do the authors believe that there is a causal relationship between partial agonism and bias?
5. It would be interesting to know how variable the responses are among the opioids. Have the authors examined other opioid drugs, such as DAMGO or methadone?

Reviewer #2 (Remarks to the Author):

Summary of Key Findings

Using a discovery-based proteomics approach, Schattauer et al., have identified a signaling enzyme peroxiredoxin 6 (PRDX6), as an opioid receptor-mediated, JNK-dependent, G α i (Gai) associated protein. PRDX6, is a dual enzyme that has separately regulated glutathione peroxidase, and phospholipase A2 (PLA2) activities. Schattauer et al., suggest that selective ligands of the kappa opioid and mu opioid receptors (KOR and MOR respectively) mediate JNK activation that results in activation of PRDX6 promoting its translocation from the cytosol to the plasma membrane where its iPLA2 activity increases and ROS production enhance opioid receptor-Gai association. Furthermore, the opioid receptor-promoted ROS reduces the palmitoylation of the receptor-associated Gai, locking the receptor and Gai in an inactive state. They also show that inhibition of PDRX6 activity in vivo blocks acute analgesic tolerance to morphine, and norBNI-mediated KOR inactivation.

Reviewer comments

Overall this is a well written manuscript that describes a series of easy to follow experiments that have yielded interesting data in support of much of the authors claims. However, there are a number of short comings that need to be addressed for consideration for publication:-

i) The proteomics data identified PDRX6 as a norBNI/KOR-dependent Gai associated protein. As these data support an association of PDRX6 with Gai rather than a direct interaction, a biochemical assay and co-immunoprecipitation using purified proteins, or FRET/BRET could be used to further interrogate the interaction to determine if the interaction is direct/indirect.

ii) The reversible palmitoylation of both GPCRs and G proteins typically triggered by ligand activation has long been recognized as one mechanism that regulates their location and activity. Ga palmitoylation has been shown to enhance Ga binding to membranes and other proteins, whereas depalmitoylation of Ga allows Ga to translocate from the membrane. Also receptor palmitoylation has been shown to be required for desensitization of certain receptors, but the mechanism regulating the palmitoylation/ depalmitoylation cycle has not been characterized. The authors present data that suggests KOR and MOR-activated JNK promotes the activation and translocation of PDRX6 from the cytosol to the plasma membrane to enhance the receptor-Gai interaction and it does so as a consequence of its PLA2 activity and ROS production that leads to a reduction in Gai palmitoylation. They also state that previous studies have shown that MAPK phosphorylates PDRX6 leading to increased membrane localization (Chatterjee et al; JBC, 2011; ref 14). Although JNK and PLA2 inhibition blocked the norBNI-induced increase in receptor-Gai association, the authors do not present corroborating evidence to demonstrate, a) JNK-dependent translocation of PDRX6, i.e., using a microscopy approach akin to tracking β arrestin-GFP translocation and, b) that PDRX6 is a JNK substrate, there is no evidence that PDRX6 phosphorylation has been investigated. A phospho-PRDX6 specific antibody is described by Chatterjee et al., (ref 14).

iii) Supplementary Figure 4E: siPRDX6 compared to siCON fluorescence shows incomplete knockdown of PRDX6 (as is often the case with siRNA knockdown), can the authors quantitate % knockdown? Moreover, can the knockdown effect be rescued by overexpressing a wobble mutant of PDRX6?

Also siPDRX6 appears to have some effect on ROS production compared to siCON transfected cells albeit that there is no enhancement of ROS in norBNI treated siPDRX6 transfected cells. Can the authors comment on this?

iv) Cross inhibition of D2DR. Here the authors attempt to extend the role of PDRX6 in GPCR desensitization to be more broadly applicable to Gai-coupled receptors and show that following agonist activation of D2DR PDRX6-PLA2 activity and ROS production is observed, and ROS production is diminished in the presence of the PDRX6 PLA2-selective inhibitor MJ33 demonstrating that the D2DR agonist-stimulated ROS production is PDRX6-dependent. However, again the phosphorylation status of PRDX6 following quinpirole activation of D2DR has not been investigated and although ROS production appears to be PRDX6-dependent, unlike KOR or MOR it is clearly not JNK-dependent but presumably quinpirole-stimulated MAPK kinase leads to phosphorylation of PDRX6. Instead, the authors demonstrate that the JNK-dependency comes from the heterologous desensitization of D2DR following KOR treatment with norBNI. A 'KOR only' control is missing from Figure 6.

Robinson et al., (Cell Signaling; 2013) previously described JNK-dependent cross-inhibition of KOR by the Orexin 1 receptor (OX1R), and it may be that OX1R-mediated JNK activation also leads to PDRX6 activation and translocation which may account for the observed attenuation of KOR Gqi activity that Robinson et al reported. It would be appropriate to cite this paper when discussing cross inhibition.

v) The statement at the beginning of the discussion is a little misleading as it states that JNK activation inhibits KOR, MOR, and D2DR by 'directly acting on the receptor signaling complexes', the data support this claim with regards to KOR and MOR but the effect is indirect for D2DR. Similarly, the SILAC data was generated using KOR, and not MOR or D2DR. Authors should rephrase these sentences to reflect the experiments that were performed.

Also in the discussion the authors state that 'a role for PRDX6 in GPCR signaling has not been previously suggested'. However, as mentioned, they cite work by Chatterjee et al; JBC, 2011, (ref 14) in reference to MAPK phosphorylation of PDRX6 (ERK1/2 or p38, but not JNK) being required for its activation and translocation to the plasma membrane where it co-localizes with an integral membrane protein. But Chatterjee et al., also show that the MAPK-dependent PRDX6 phosphorylation and translocation to the plasma membrane and subsequent ROS production was mediated by Ang II activation of the Angiotensin 1 receptor which implicates a role for PRDX6 in AT1-R, (a Gαq-coupled GPCR) signaling. So, in addition to citing Chatterjee et al., for demonstrating MAPK phosphorylation of PDRX6, the authors should also mention and cite where appropriate, the studies describing AT-1R-mediated phosphorylation and translocation of PDRX6.

vi) As mentioned, the authors suggest that the PRDX6-dependent receptor desensitization mechanism may go beyond opioid receptors and be more applicable to Gai-coupled receptors in general. So they extended their studies to include D2DR and show quinpirole activated D2DR activity is also regulated by PRDX6. However, they have not ruled out the mechanism being extended to Gas or Gαq-coupled receptors. It would be interesting and helpful to include examples of Gas and Gαq coupled receptors to investigate whether this PRDX6-dependent receptor desensitization mechanism is exclusive to Gai-coupled receptors.

vii) In vivo studies. Following on from their observations in vitro, Schattauer et al., next investigated whether these observations have physiological relevance. The in vivo tolerance studies indicate that KOR-mediated JNK activation does lead to cross inhibition of the D2DR in vivo. However, as the authors suggest PDRX6 regulation of GPCR activity may be a general regulatory mechanism that can also promote cross inhibition (heterologous desensitization) of other neighboring GPCRs. In addition to D2DR, it would be interesting and add more weight to the authors claims if cross inhibition of other GPCRs was investigated both in vitro and in vivo.

Minor concerns

1. The authors draw an analogy between the effects of PDRX6 in attenuating receptor signaling to that of 'arrestin', a well-known regulator of GPCR activity. They refer to PDRX6 as being 'arrestin-like'. As arrestin is a multi-faceted protein with a well-known role in receptor desensitization but in recent years it has been demonstrated that arrestin serves many other functions, including that as a MAPK module scaffold. I would suggest that the authors re-phrase 'arrestin-like' and refer to PDRX6 as having 'arrestin-like' attributes with regards to its role in receptor desensitization or something along these lines.

2. Some of the figures related to immunoprecipitation experiments are presented as box graphs and others as bar graphs, Are two methods of graphing necessary?

3. Some graphs represent data as '% of vehicle', but vehicle is either not shown or not set as 100%.

4. The authors suggest that PDRX6 inhibitors may have therapeutic utility as adjuncts to opioids to reduce the development of tolerance; it should be noted that PRDX6 knock-out mice exhibit low survival rates, severe tissue damage and high protein oxidation levels suggesting PDRX6 inhibitors are unlikely to be useful clinically.

In conclusion, arguably, a role for PDRX6 in GPCR signaling is not novel however, the novelty

herein lies in defining its role and identifying the selective PLA2 activity of PDRX6 as enhancing opioid receptor and Gai association and that this enhanced association attenuates palmitoylation of the Gai protein, locking the receptor/Gai complex in an inactive state. Moreover, this provides a potential explanation for the prolonged antagonist effect observed for some selective KOR antagonists such as norBNI. Based on the data presented in the Schattauer manuscript together with the reports of Chatterjee (2011) and Robinson (2013), it is plausible that MAPK-dependent PRDX6 regulation of GPCR signaling is an even more general mechanism that spans across other G protein classes and involves other MAPK family members. That said, the data presented by Schattauer is very interesting in that it provides a framework and evidence for a novel arrestin-independent, GPCR desensitization mechanism, and the authors have done a good job pulling together pieces of a puzzle that has eluded researchers in the opioid field for many years (i.e. how JNK activation by selective, long-acting KOR antagonists results in prolonged receptor inactivation). The cross inhibition angle is also interesting, however, the manuscript would benefit from focusing on the opioid receptors (KOR and MOR) and addressing some of the fundamental experiments that are lacking and that will strengthen their conclusions, and perhaps follow up with the cross inhibition story.

Reviewer #3 (Remarks to the Author):

Schattauer et al. describes the recruitment of peroxiredoxin 6 (PRDX6) to the opioid receptor complex. Next, the functional relevance of this interaction is studied.

In my opinion the experiments on their own are performed in a proper way, including necessary controls. However, I'm a bit confused by the interpretation and the conclusions made out of these results, especially the involvement of PRDX6 is often shown in an indirect way e.g. by the use of MJ33 (an inhibitor of PLA2).

Figure 2 shows the interaction between Gai and PRDX6 (and KOR) upon receptor stimulation. From these data I'm not convinced that PRDX6 regulates the association between KOR and Gai. To have more evidence a siRNA(PRDX6) experiment can be performed.

Figure 3 shows a decrease in Gai palmitoylation upon stimulation of KOR (by norBNI and morphine). Also here there is no direct evidence for the role of PRDX6.

Can you give some extra explanation, rationale, experiments,... to convince me this is a correct approach. This is also necessary to make conclusions out of the in vivo data and the D2R data.

Some more details can be included about:

"unexplained pharmacological properties"

"ATF-2, ATF-3"

norBNI is described to be an antagonist but from the results shown in this manuscript it seems to be an agonist. Can you give some details about this.

Figure 7 is a bit confusing as from the data there is no evidence for a direct interaction of PRDX6 with the betagamma subunits of the G protein.

Legend of suppl. fig 2: add an explanation for ROK0, etc. (see M&M)

Pay some more attention to the lay out of the figures. Often the font is squeezed or the '-' and '+' are dancing.

Some minor typo's:

p12: myristoylation: add 'y'

p21: ... or saline): add ')''

p22: correct PRDX6-dependent

p23: add antibody: ...IRDye secondary

Reviewer #1 (Remarks to the Author):

This submission examines a novel pathway involved with inactivation/inhibition of GPCR responses. It is based upon earlier work from the same laboratory indicating that the prolonged inhibition of kappa actions by antagonists such as norBNI result from activation of JNK. The current submission extends this to provide a mechanism whereby that results in formation of the generation of reactive oxygen species produced by perioxiredoxin6 (PRDX6), which in turn results in the depalmitoylation of G α i. The work is carefully carried out with the appropriate controls and presents a logical extension of the earlier findings. The authors extend this to show evidence of heterologous desensitization between D2 dopamine receptors and opioid receptors. Technically, the work is well described and the conclusions supported by the evidence.

Some issues that require some discussion:

1. A major question involves why a kappa antagonist and a mu agonist elicit the same response. The actions of norBNI would suggest that the pathway is not initiated by activation of the receptor. How do they envisage this response being initiated? Is it possible that norBNI is interacting with an allosteric site or even an alternative target? It would be helpful to the reader to have a better understanding of the mechanisms involved if they are known.

Authors' Response: The crystal structure of JDTC bound to the kappa receptor published by Ray Stevens' group (Wu et al., Nature 2012) indicates that the long-lasting antagonists including norBNI can bind at the same pocket as the agonists do. While it is possible that the antagonists bind to an additional allosteric site, there is currently no evidence for that. Instead we believe that norBNI is a novel type of biased agonist selective for the JNK/PRDX6 signaling pathway. The current study is focused on the consequences downstream of JNK activation. It is possible that the list of interacting proteins will provide insights for future work on this issue. We have clarified in the introduction that prior studies have demonstrated that norBNI-stimulation of JNK is KOR-dependent, and added to the discussion that this is a remaining question.

2. In Fig. 4c, it appears that the peak MJ33 effect is similar, but it appears to diminish more slowly. It would help if the authors determined the area under the curve (AUC) to compare the full effect of the drug rather than just peak effect

Authors' Response: The AUC for the effect of MJ33 on morphine (Fig. 4c) and fentanyl (Fig. 4d) -induced analgesia is provided in Supplemental Figures 7b and 7b, respectively. We have edited the figure legend for Fig. 4 to more clearly reference this.

3. The authors examine 'acute tolerance'. While this is a well established entity, it is not clear that it has any relationship to the tolerance produced by treating subjects for weeks to months, which is the clinical situation. The authors should discuss this and, if possible, look at the effects of chronic morphine to see if they replicate those seen acutely.

Authors' Response: As requested, we have added an experiment looking at the effects of MJ33 on the antinociceptive responses to repeated morphine (Fig. 4e), which demonstrates the MJ33 delays tolerance to morphine (5 mg/kg, twice a day) by 4 days. Tolerance to morphine is known to be mediated by multiple adaptive mechanisms.

4. The authors point out the difference between fentanyl and morphine. While they allude to possible differences in B-arrestin signaling and partial agonists, this should be explained in greater detail. How do the biased factors of the compounds compare? Do the authors believe that there is a causal relationship between partial agonism and bias?

Authors' Response: The relationship between ligand bias and agonist efficacy as classically defined is still being experimentally resolved. We know that GRK activation requires Gbg activation, and it has been established that partial agonists including morphine are less effective at arrestin recruitment. The dominant hypothesis in the field is that ligand binding stabilizes different poses of the receptor that result in distinct forms of signaling, but these conformations have not yet been structurally established. We have elaborated on this in the discussion. We have also distinguished that partial agonists and biased agonists may be distinct in some cases, but both could contribute to low efficacy for arrestin recruitment and the ability to stimulate PRDX6/ROS-dependent signaling and desensitization.

5. It would be interesting to know how variable the responses are among the opioids. Have the authors examined other opioid drugs, such as DAMGO or methadone?

Authors' Response: We have not yet addressed this interesting question. However, based on the differences observed between morphine and fentanyl, we would predict that opioid responses would be related to efficacy of arrestin recruitment. In the discussion, we acknowledge that further work will be needed to look at the variability in responses among opioids.

Reviewer #2 (Remarks to the Author):

Summary of Key Findings: Using a discovery-based proteomics approach, Schattauer et al., have identified a signaling enzyme peroxidase 6 (PRDX6), as an opioid receptor-mediated, JNK-dependent, G α i associated protein. PRDX6, is a dual enzyme that has separately regulated glutathione peroxidase, and phospholipase A2 (PLA2) activities. Schattauer et al., suggest that selective ligands of the kappa opioid and mu opioid receptors (KOR and MOR respectively) mediate JNK activation that results in activation of PRDX6 promoting its translocation from the cytosol to the plasma membrane where its iPLA2 activity increases and ROS production enhance opioid receptor-G α i association. Furthermore, the opioid receptor-promoted ROS reduces the palmitoylation of the receptor-associated G α i, locking the receptor and G α i in an inactive state. They also show that inhibition of PDRX6 activity in vivo blocks acute analgesic tolerance to morphine, and norBNI-mediated KOR inactivation.

Reviewer comments: Overall this is a well written manuscript that describes a series of easy to follow experiments that have yielded interesting data in support of much of the authors

claims. However, there are a number of shortcomings that need to be addressed for consideration for publication.

i) The proteomics data identified PDRX6 as a norBNI/KOR-dependent G α i associated protein. As these data support an association of PDRX6 with G α i rather than a direct interaction, a biochemical assay and co-immunoprecipitation using purified proteins, or FRET/BRET could be used to further interrogate the interaction to determine if the interaction is direct/indirect.

Authors' Response: We have changed the text to address that our current data do not determine if the interaction is direct or involves intermediates. Developing a new FRET/BRET or purified protein system with G α i/PRDX6 is something we are interested in pursuing in the future, but would be technically challenging and beyond the scope of the current paper.

ii) The reversible palmitoylation of both GPCRs and G proteins typically triggered by ligand activation has long been recognized as one mechanism that regulates their location and activity. G α palmitoylation has been shown to enhance G α binding to membranes and other proteins, whereas depalmitoylation of G α allows G α to translocate from the membrane. Also receptor palmitoylation has been shown to be required for desensitization of certain receptors, but the mechanism regulating the palmitoylation/ depalmitoylation cycle has not been characterized. The authors present data that suggests KOR and MOR-activated JNK promotes the activation and translocation of PDRX6 from the cytosol to the plasma membrane to enhance the receptor-G α i interaction and it does so as a consequence of its PLA2 activity and ROS production that leads to a reduction in G α i palmitoylation. They also state that previous studies have shown that MAPK phosphorylates PDRX6 leading to increased membrane localization (Chatterjee et al; JBC, 2011; ref 14). Although JNK and PLA2 inhibition blocked the norBNI-induced increase in receptor-G α i association, the authors do not present corroborating evidence to demonstrate, a) JNK-dependent translocation of PDRX6, i.e., using a microscopy approach akin to tracking β arrestin-GFP translocation and, b) that PDRX6 is a JNK substrate, there is no evidence that PDRX6 phosphorylation has been investigated. A phospho-PRDX6 specific antibody is described by Chatterjee et al., (ref 14).

Authors' Response: As requested, we have now added data demonstrating JNK-dependent PRDX6 translocation by microscopy and by western analysis of cellular fractions (Fig. 2D-F; Supplementary Fig 4). This enhances the conclusions drawn from the presence of PRDX6-PLA2 activity only in the membrane fraction of cells. We have also added to the text that we have been unable to detect PRDX6 phosphorylation by phosphoproteomic approaches.

iii) Supplementary Figure 4E: siPRDX6 compared to siCON fluorescence shows incomplete knockdown of PRDX6 (as is often the case with siRNA knockdown), can the authors quantitate % knockdown? Moreover, can the knockdown effect be rescued by overexpressing a wobble mutant of PDRX6?

Authors' Response: The % knockdown, as measured by PRDX6 immunofluorescence, is now presented in the figure legend of Supplementary Fig 3. We have also added to the methods section the % knockdown observed by western analysis when selecting siRNA transfection conditions. We have not designed a

PRDX6 construct with wobble mutations at both siRNAs used in the knockdown experiments.

Also, siPDRX6 appears to have some effect on ROS production compared to siCON transfected cells albeit that there is no enhancement of ROS in norBNI treated siPDRX6 transfected cells. Can the authors comment on this?

Authors' Response: No statistical difference was observed between siPRDX6 and siCON transfected cells under vehicle treated conditions. CellRox Green fluorescence in vehicle treated siPRDX6 transfected cells was measured to be 123±31% of control. We have added this information to the figure legend of Supplementary Fig. 5, and replaced these images in Supplementary Fig. 5e with more representative images.

iv) Cross inhibition of D2DR. Here the authors attempt to extend the role of PDRX6 in GPCR desensitization to be more broadly applicable to G α i-coupled receptors and show that following agonist activation of D2DR PDRX6-PLA2 activity and ROS production is observed, and ROS production is diminished in the presence of the PDRX6 PLA2-selective inhibitor MJ33 demonstrating that the D2DR agonist-stimulated ROS production is PDRX6-dependent. However, again the phosphorylation status of PRDX6 following quinpirole activation of D2DR has not been investigated and although ROS production appears to be PRDX6-dependent, unlike KOR or MOR it is clearly not JNK-dependent but presumably quinprinine-stimulated MAPK kinase leads to phosphorylation of PDRX6. Instead, the authors demonstrate that the JNK-dependency comes from the heterologous desensitization of D2DR following KOR treatment with norBNI. A 'KOR only' control is missing from Figure 6.

Authors' Response: We have clarified that our studies show a role for PRDX6-dependent ROS production and tolerance in D2DR (but not necessarily JNK). We have also added the "KOR only" control (Supplementary figure 8e).

Robinson et al., (Cell Signaling; 2013) previously described JNK-dependent cross-inhibition of KOR by the Orexin 1 receptor (OX1R), and it may be that OX1R-mediated JNK activation also leads to PDRX6 activation and translocation which may account for the observed attenuation of KOR G α i activity that Robinson et al reported. It would be appropriate to cite this paper when discussing cross inhibition.

Authors' Response: We have added this reference to the discussion when discussing the cross-inhibition.

v) The statement at the beginning of the discussion is a little misleading as it states that JNK activation inhibits KOR, MOR, and D2DR by 'directly acting on the receptor signaling complexes', the data support this claim with regards to KOR and MOR but the effect is indirect for D2DR. Similarly, the SILAC data was generated using KOR, and not MOR or D2DR. Authors should re-phrase these sentences to reflect the experiments that were performed.

Authors' Response: We have edited the text to more accurately reflect this distinction.

Also in the discussion the authors state that 'a role for PRDX6 in GPCR signaling has not been previously suggested'. However, as mentioned, they cite work by Chatterjee et al; JBC, 2011, (ref 14) in reference to MAPK phosphorylation of PDRX6 (ERK1/2 or p38, but not JNK) being required for its activation and translocation to the plasma membrane where it co-localizes with an integral membrane protein. But Chatterjee et al., also show that the MAPK-dependent PRDX6 phosphorylation and translocation to the plasma membrane and subsequent ROS production was mediated by Ang II activation of the Angiotensin 1 receptor which implicates a role for PRDX6 in AT1-R, (a Gαq-coupled GPCR) signaling. So, in addition to citing Chatterjee et al., for demonstrating MAPK phosphorylation of PDRX6, the authors should also mention and cite where appropriate, the studies describing AT-1R-mediated phosphorylation and translocation of PDRX6.

Authors' Response: We have added angiotensin receptor mediated activation to the relevant section, and altered the discussion statement mentioned to more accurately reflect that the angiotensin receptor was previously implicated.

vi) As mentioned, the authors suggest that the PRDX6-dependent receptor desensitization mechanism may go beyond opioid receptors and be more applicable to Gai-coupled receptors in general. So they extended their studies to include D2DR and show quinpirole activated D2DR activity is also regulated by PRDX6. However, they have not ruled out the mechanism being extended to Gas or Gαq-coupled receptors. It would be interesting and helpful to include examples of Gas and Gαq coupled receptors to investigate whether this PRDX6-dependent receptor desensitization mechanism is exclusive to Gai-coupled receptors.

Authors' Response: The question of if this desensitization mechanism also regulates Gas and Gαq is an interesting question we plan to pursue in other studies, but we feel would distract from the main focus on a novel desensitization mechanism. We have changed the language in the discussion to address this uncertainty.

vii) In vivo studies. Following on from their observations in vitro, Schattauer et al., next investigated whether these observations have physiological relevance. The in vivo tolerance studies indicate that KOR-mediated JNK activation does lead to cross inhibition of the D2DR in vivo. However, as the authors suggest PDRX6 regulation of GPCR activity may be a general regulatory mechanism that can also promote cross inhibition (heterologous desensitization) of other neighboring GPCRs. In addition to D2DR, it would be interesting and add more weight to the authors claims if cross inhibition of other GPCRs was investigated both in vitro and in vivo.

Authors' Response: Similar to the question of other classes of GPCRs, this is a question we are interested in pursuing in future studies, but we agree with the reviewers later comments that this manuscript would be clearest if primarily focused on opioid receptors. Our data however demonstrate the possibility of cross-desensitization, which is a valuable insight. We have added to the discussion that it is not clear how common this cross inhibition is to address this concern.

Minor concerns

1. The authors draw an analogy between the effects of PDRX6 in attenuating receptor signaling to that of 'arrestin', a well-known regulator of GPCR activity. They refer to PDRX6 as being 'arrestin-like'. As arrestin is a multi-faceted protein with a well-known role in receptor desensitization but in recent years it has been demonstrated that arrestin serves many other functions, including that as a MAPK module scaffold. I would suggest that the authors re-phrase 'arrestin-like' and refer to PDRX6 as having 'arrestin-like' attributes with regards to its role in receptor desensitization or something along these lines.

Authors' Response: We have edited the text to address this.

2. Some of the figures related to immunoprecipitation experiments are presented as box graphs and others as bar graphs, Are two methods of graphing necessary?

Authors' Response: All of the box graphs have been converted to bar graphs.

3. Some graphs represent data as '% of vehicle', but vehicle is either not shown or not set as 100%.

Authors' Response: Supplementary Fig. 4C (now 5C) was missing vehicle control images, and the normalization in Supplementary Fig 4D (now 5D) was unclearly described. These images have been added, and the appropriate description added to the figure legend. Figures 5C,D, did not originally show the saline-baseline (quinpirole) locomotor group, which has been added. We apologize for these omissions. Elsewhere, when vehicle is not shown, vehicle was defined as 100% and experimental groups were normalized to vehicle in individual experiments.

4. The authors suggest that PDRX6 inhibitors may have therapeutic utility as adjuncts to opioids to reduce the development of tolerance; it should be noted that PRDX6 knock-out mice exhibit low survival rates, severe tissue damage and high protein oxidation levels suggesting PDRX6 inhibitors are unlikely to be useful clinically.

Authors' Response: PRDX6 knockout mice are reported to develop normally but have elevated hydrogen peroxide and protein oxidation levels. However, they show vulnerability to oxidative stress, after which low survival rates and severe tissue damage is observed (Wang, et.al., JBC 2003). Additionally, while PRDX6 knockout removes both the peroxidase and phospholipase enzyme function of PRDX6, MJ33 inhibits the phospholipase activity with minimal effect on PRDX6 peroxidase activity. A reference to a study demonstrating low side effects following 4 days of MJ33 administration has been added. (Lee, et.al., JPET 2013). We have also added a study (Fig. 4E) demonstrating MJ33 delays the development of tolerance to chronic morphine; no adverse effects to MJ33 were observed during this study.

In conclusion, arguably, a role for PDRX6 in GPCR signaling is not novel however, the novelty herein lies in defining its role and identifying the selective PLA2 activity of PDRX6 as enhancing opioid receptor and Gai association and that this enhanced association attenuates palmitoylation of the Gai protein, locking the receptor/Gai complex in an inactive state. Moreover, this provides a potential explanation for the prolonged antagonist effect observed for some selective KOR antagonists such as norBNI. Based on the data presented in

the Schattauer manuscript together with the reports of Chatterjee (2011) and Robinson (2013), it is plausible that MAPK-dependent PRDX6 regulation of GPCR signaling is an even more general mechanism that spans across other G protein classes and involves other MAPK family members. That said, the data presented by Schattauer is very interesting in that it provides a framework and evidence for a novel arrestin-independent, GPCR desensitization mechanism, and the authors have done a good job pulling together pieces of a puzzle that has eluded researchers in the opioid field for many years (i.e how JNK activation by selective, long-acting KOR antagonists results in prolonged receptor inactivation). The cross inhibition angle is also interesting, however, the manuscript would benefit from focusing on the opioid receptors (KOR and MOR) and addressing some of the fundamental experiments that are lacking and that will strengthen their conclusions, and perhaps follow up with the cross inhibition story.

Reviewer #3 (Remarks to the Author):

Schattauer et al. describes the recruitment of peroxiredoxin 6 (PRDX6) to the opioid receptor complex. Next, the functional relevance of this interaction is studied. In my opinion the experiments on their own are performed in a proper way, including necessary controls. However, I'm a bit confused by the interpretation and the conclusions made out of these results, especially the involvement of PRDX6 is often shown in an indirect way e.g. by the use of MJ33 (an inhibitor of PLA2). Figure 2 shows the interaction between Gai and PRDX6 (and KOR) upon receptor stimulation. From these data I'm not convinced that PRDX6 regulates the association between KOR and Gai. To have more evidence a siRNA(PRDX6) experiment can be performed. Figure 3 shows a decrease in Gai palmitoylation upon stimulation of KOR (by norBNI and morphine). Also here there is no direct evidence for the role of PRDX6. Can you give some extra explanation, rationale, experiments,... to convince me this is a correct approach. This is also necessary to make conclusions out of the in vivo data and the D2R data.

Authors' Response: We have clarified (with appropriate citations) in the text that MJ33 is not a broad spectrum PLA2 inhibitor, but is selective for pancreatic PLA2 and PRDX6. In addition to our data demonstrating calcium-dependent MJ33-sensitive PLA2 activity in the membrane fraction (Fig 2E), we have added experiments demonstrating PRDX6 translocates to the membrane following norBNI stimulation (Fig 2F,G). These data combined should more conclusively demonstrate the role of PRDX6 in norBNI induced PLA2 activity.

While siRNA was not used in all experiments, siRNA was used to show PRDX6 mediated the stimulation of reactive oxygen species by norBNI. Our argument is further strengthened by the lack of effect of a different PLA2 inhibitor (AACOCF3), which targets multiple PLA2 enzymes groups but has minimal efficacy at PRDX6. This inhibitor was used in the reactive oxygen species experiments (Fig 3) as well as the in vivo experiments (Fig 4A,C).

Some more details can be included about:

“unexplained pharmacological properties”

Authors' Response: We have edited this sentence to refer to the long duration of action, which is further discussed in the following paragraph.

“ATF-2, ATF-3”

ATF-2 was used as a positive control to validate JNK kinase activity was observed under our conditions. This has been clarified in the methods section.

norBNI is described to be an antagonist but from the results shown in this manuscript it seems to be an agonist. Can you give some details about this.

In the discussion of norBNI-like KOR antagonists in the second paragraph, we added text describing norBNI as having collateral agonist activity. This sentence described that despite generally being considered antagonists, they activate the JNK pathway. We hope this is clearer.

Figure 7 is a bit confusing as from the data there is no evidence for a direct interaction of PRDX6 with the betagamma subunits of the G protein.

Authors' Response: We thank the reviewer for noticing this misleading detail. Figure 7c has been corrected so that it doesn't imply an interaction between G $\beta\gamma$ and PRDX6.

Legend of suppl. fig 2: add an explanation for R0K0, etc. (see M&M)

Authors' Response: This information has been added to the figure legend, and the abbreviation used within Supplementary Fig. 2 have been added to the methods.

Pay some more attention to the lay out of the figures. Often the font is squeezed or the ‘-’ and ‘+’ are dancing.

Some minor typo's:

p12: myristoylation: add ‘y’

p21: ... or saline): add ‘)’

p22: correct PRDX6-dependent

p23: add antibody: ...IRDye secondary

Authors' Response: We thank the reviewer for identifying these issues, which we have corrected.

REVIEWERS' COMMENTS:

Reviewer #1 (Remarks to the Author):

My concerns have been adequately addressed.

Reviewer #2 (Remarks to the Author):

The authors have responded adequately to previous concerns with new experimental work, including data demonstrating JNK-dependent PRDX6 translocation using microscopy and western blot analysis of cellular fractions. The new data and additional comments in the discussion are consistent with and further support the previous conclusions that selective activation of opioid receptors leads to JNK-mediated activation of PRDX6 promoting its translocation from the cytosol to the PM where its iPLA2 activity increases ROS production enhancing opioid receptor-Gai association. This work is an interesting and important contribution to the field and is now suitable for publication

Reviewer #3 only commented privately to the editor and stated that hers/his previous concerns were addressed.

Reviewer #4 (Remarks to the Author):

This manuscript proposes that the PLA2 activity of peroxiredoxin 6 (Prdx6) results in the production of reactive oxygen species (ROS) that in turn results in lipid oxidation and inactivation of the opioid receptor. This is a novel hypothesis of potentially great clinical importance as pointed out in the manuscript. Opioid receptor activity is crucial to the understanding of opioid tolerance and its manipulation may provide the methodology to modulate tolerance and its side effects. The manuscript presents significant support for the hypothesis. This evidence includes demonstration of Prdx6 association that is enhanced by norBNI treatment, inhibition of the effect by the relatively specific Prdx6 PLA2 inhibitor MJ33, the demonstration of ROS production in response to opioids and the demonstration that Prdx6 in vivo mediates opioid receptor inactivation. While the basic premise appears to be well substantiated by the data, the evidence for the proposed mechanism is not as strong. This is indicated by the following points:

1) The failure to detect phosphorylated Prdx6 is somewhat disconcerting as this mechanism is necessary for activation of the PLA2 activity. Further, previous studies have indicated Prdx6 phosphorylation by Erk and p38 but not JNK. On the other hand, localization of the activity to the membrane fraction and its inhibition by MJ33 provide support for Prdx6 PLA2 activation. As pointed out in the manuscript, there may be cell and tissue differences in the response to the various kinases. It is possible that a relatively small fraction of Prdx6 was phosphorylated but that was insufficient for detection by MS. Perhaps western blot would be more successful in detecting some degree of protein phosphorylation (we would be happy to provide the antibody to phosphorylated Prdx6). Another test could be inhibition of the in vivo effects with an upstream inhibitor of MAPK activity.

2) The PLA2 inhibitor studies provides good evidence for the role of Prdx6 in vivo. However, this aspect of this study would be enhanced through the use of Prdx6 null mice (readily available from JAX) or the use of D140A-Prdx6 mice that express only the peroxidase but not the PLA2 activity of Prdx6 (we will be happy to supply those mice).

3) the manuscript points out that Prdx6 can activate NADPH oxidase type (NOX2) and this is the proposed source of ROS. However, this specific source of ROS is not specifically evaluated. NOX2 null mice are readily available and could be used to test for the source of ROS. Alternatively, the mechanism for Prdx6 activation of NOX2 is through activation of rac and one of the several

inhibitors of this pathway could be tested for its effect (J. Vazquez Medina et al, FASEB J 30:2885, 2016).

Minor Comments:

- 1) The abstract states that Prdx6 generates ROS, which it does but indirectly. I suggest adding "...via activation of NADPH oxidase..." to that sentence.
- 2) line 936, Ref. 48 is indicated but not given in the ref list. I suppose that the ref. is Lee. he et al, JPET 345:284, 2013.
- 3) line 846, please state the pH used for the assay. It would be useful to state the absolute PLA2 activity in the text; Fig.2i gives relative changes.

Reviewers #1-3 had no residual concerns.

Reviewer #4 (Remarks to the Author):

This manuscript proposes that the PLA2 activity of peroxiredoxin 6 (Prdx6) results in the production of reactive oxygen species (ROS) that in turn results in lipid oxidation and inactivation of the opioid receptor. This is a novel hypothesis of potentially great clinical importance as pointed out in the manuscript. Opioid receptor activity is crucial to the understanding of opioid tolerance and its manipulation may provide the methodology to modulate tolerance and its side effects. The manuscript presents significant support for the hypothesis. This evidence includes demonstration of Prdx6 association that is enhanced by norBNI treatment, inhibition of the effect by the relatively specific Prdx6 PLA2 inhibitor MJ33, the demonstration of ROS production in response to opioids and the demonstration that Prdx6 in vivo mediates opioid receptor inactivation. While the basic premise appears to be well substantiated by the data, the evidence for the proposed mechanism is not as strong. This is indicated by the following points:

1) The failure to detect phosphorylated Prdx6 is somewhat disconcerting as this mechanism is necessary for activation of the PLA2 activity. Further, previous studies have indicated Prdx6 phosphorylation by Erk and p38 but not JNK. On the other hand, localization of the activity to the membrane fraction and its inhibition by MJ33 provide support for Prdx6 PLA2 activation. As pointed out in the manuscript, there may be cell and tissue differences in the response to the various kinases. It is possible that a relatively small fraction of Prdx6 was phosphorylated but that was insufficient for detection by MS. Perhaps western blot would be more successful in detecting some degree of protein phosphorylation (we would be happy to provide the antibody to phosphorylated Prdx6). Another test could be inhibition of the in vivo effects with an upstream inhibitor of MAPK activity.

Authors' response: We are well aware of the studies defining the mechanism of ERK activation of PRDX6 that the reviewer is listing, and we cite them in the manuscript. We looked closely for phosphopeptides by mass spec as we described. While we see other phosphopeptides in the assay, we did not find the predicted phospho-PRDX6 peptides, although the non-phosphopeptides were detected. We have added a statement that while our data suggest JNK activation of PRDX6 function may be involve a different mechanism (perhaps phosphorylation of a chaperone mediating translocation), it is also possible that phosphorylated Prdx6 levels were below the threshold for detection in phospho-proteomics. We did obtain a sample of the phospho antibody from Dr. Aron Fisher's lab. The western blots had numerous nonspecific bands, but were not affected by JNK activation.

We have also clarified in the discussion that prior studies have shown that morphine tolerance and the duration of norBNI are specifically dependent on JNK MAPK (references 4-6, 10). It is also worth noting that while norBNI stimulates JNK phosphorylation, it has no agonist activity towards the p38 or ERK1/2 pathways, and thus these pathways cannot mediate the effect of norBNI on PRDX6.

2) The PLA2 inhibitor studies provides good evidence for the role of Prdx6 in vivo. However, this aspect of this study would be enhanced through the use of Prdx6 null mice (readily available from JAX) or the

use of D140A-Prdx6 mice that express only the peroxidase but not the PLA2 activity of Prdx6 (we will be happy to supply those mice).

Authors' response: We have added a note to the discussion (with citation) that the D140A-Prdx6 knock in mice would be valuable for further validation. Including these transgenic mice in the present manuscript (including breeding, validating and analyzing) seems unnecessary to us and would expand the scope beyond what could fit reasonably in one paper.

3) the manuscript points out that Prdx6 can activate NADPH oxidase type (NOX2) and this is the proposed source of ROS. However, this specific source of ROS is not specifically evaluated. NOX2 null mice are readily available and could be used to test for the source of ROS. Alternatively, the mechanism for Prdx6 activation of NOX2 is through activation of rac and one of the several inhibitors of this pathway could be tested for its effect (J. Vazquez Medina et al, FASEB J 30:2885, 2016).

Authors' response: Similar to concern 2, we would like to use the NOX2 null mice in future studies, but the time required would not be feasible for the current study. Unfortunately, as inhibitors of rac are likely to also affect MAPK activation, in addition to NADPH oxidase complex formation, experiments using these inhibitors would be challenging to draw conclusions from. We have cited other studies that have implicated NADPH oxidase subunits in morphine tolerance however (references 28 and 29), and we have added a note to the discussion that these mice would be useful for further validation.

Minor Comments:

1) The abstract states that Prdx6 generates ROS, which it does but indirectly. I suggest adding "...via activation of NADPH oxidase..." to that sentence.

Authors' response: We have added this clarification to the abstract.

2) line 936, Ref. 48 is indicated but not given in the ref list. I suppose that the ref. is Lee. he et al, JPET 345:284, 2013.

Authors' response: The reference has been corrected.

3) line 846, please state the pH used for the assay. It would be useful to state the absolute PLA2 activity in the text; Fig.2i gives relative changes.

Authors' response: We have clarified that the assay is performed at pH 7.0 in the absence of calcium, which was previously only stated in the lysis buffer description.